# TOPOLOGICAL EXPERIENCE REPLAY

**Zhang-Wei Hong**[1]**, Tao Chen**[1]**, Yen-Chen Lin**[1]**, Joni Pajarinen**[2]**, & Pulkit Agrawal**[1]
Improbable AI Lab, Massachusetts Institute of Technology[1]
Aalto University[2]

## ABSTRACT

State-of-the-art deep $Q$-learning methods update $Q$-values using state transition tuples sampled from the experience replay buffer. This strategy often uniformly and randomly samples or prioritizes data sampling based on measures such as the temporal difference (TD) error. Such sampling strategies can be inefficient at learning $Q$-function because a state's $Q$-value depends on the $Q$-value of successor states. If the data sampling strategy ignores the precision of $Q$-value estimate of the next state, it can lead to useless and often incorrect updates to the $Q$-values. To mitigate this issue, we organize the agent's experience into a graph that explicitly tracks the dependency between $Q$-values of states. Each edge in the graph represents a transition between two states by executing a single action. We perform value backups via a breadth-first search starting from the set of terminal states and successively moving backwards. We empirically show that our method is substantially more data-efficient than several baselines on a diverse range of goal-reaching tasks. Notably, the proposed method also outperforms baselines that consume more batches of training experience and operates from high-dimensional observational data such as images.

## 1 INTRODUCTION

A significant challenge in reinforcement learning (RL) is to overcome the need for large amounts of data. Off-policy algorithms have received great attention because of its ability to reuse data by experience replay (Lin, 1992) for learning the policy. The key ingredient in off-policy methods is learning a Q-function (Watkins & Dayan, 1992) which estimates the expected sum of future rewards (i.e., return) at a state. State-of-the-art deep RL algorithms train the $Q$-function by *bootstrapping* $Q$-values from state transitions sampled from the experience replay buffer (Mnih et al., 2015). In bootstrapping, the sum of the immediate reward and the succeeding state's $Q$-value are used as the target (or *"labels"*) for the current state's $Q$-value. Because of the dependency on the successor states' $Q$-value, the order in which states are sampled to update the $Q$-value substantially influences the convergence speed of $Q$-value. An improper update order results in slow convergence.

As a motivating example, consider the Markov Decision Process (MDP) shown in Figure 1. Let the agent receive a positive reward when it reaches the goal state (labeled as G), but zero at other stated. (note that our method does not need rewards to be $0$ or $1$). Starting from state $C$, the agent can obtain the reward by visiting states in the sequence $C \rightarrow D \rightarrow G$. Now if the $Q$-value of taking action $a$ at state $D$, $Q(D, a)$, is inaccurate, then using it to update $Q(C, a)$ will lead to an incorrect estimate of $Q(C, a)$. It is therefore natural to start at the terminal state $G$ and first bootstrap $Q$-values of preceding states $(D, E)$, and then for states $(A, B, C)$. Using such a reverse order for bootstrapping ensures that for each state, the $Q$-values of successor states are updated before an update is made to the current state's value. In fact, it has been proven that for acyclic MDPs such a *reverse sweep* is the optimal order for bootstrapping (Bertsekas et al., 2000). Other works (Grześ & Hoey, 2013; Dai & Hansen, 2007; Dai et al., 2011) have further empirically demonstrated the effectiveness of *reverse sweep* in cyclical MDPs.

However, it is challenging to perform $Q$-learning with *reverse sweep* in high-dimensional state spaces since the predecessors of each state are often unknown. State-of-the-art $Q$-learning methods (Mnih et al., 2015; Lillicrap et al., 2015; Haarnoja et al., 2018; Fujimoto et al., 2018) resort to random sampling of data from the replay buffer. Their speed of convergence can be slow as these methods do not account for the structure in state transitions when selecting the order of states for

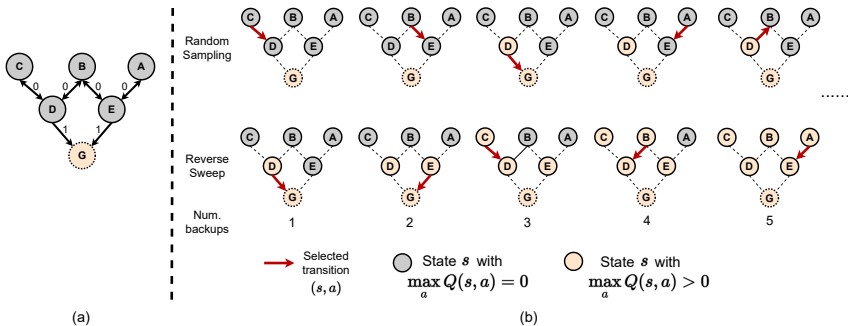

Figure 1: The ordering of states used for updating $Q$-values directly effects the convergence speed. **(a)** Consider the graphical representation of the MDP with the goal state (G). Each node in the graph is a state. The arrows denote possible transitions $(s, a, r, s')$ and the numbers on the arrows are the rewards $r$ associated with the transition. **(b)** Random sampling wastes many backups at states with zero values (gray) while each backup of reverse sweep propagates values (orange) one step back.

updating the $Q$-values. The speed of convergence matters because data collection is interleaved with $Q$-value updates. If the $Q$-values are incorrect, the agent may take actions that result in low rewards. Such data is useless for improving the $Q$-values of states in the high-return trajectories. Therefore, the slower convergence of $Q$-values is directly linked to the data inefficiency. One way to reduce dependence on data is to artificially increase the speed of convergence by increasing the ratio between the $Q$-learning update steps and the data collection steps. However, when function approximators are used to estimate the $Q$-value, excessive updates on a fixed set of interaction data can lead to over-fitting and overestimation of the $Q$-values, resulting in worse overall performance. Our experiments in Section 5.4 empirically confirm this hypothesis.

To speed up $Q$-function convergence in high-dimensional state spaces, we approximate the *reverse sweep* update scheme via stitching the trajectories stored in the replay buffer to construct a graph. The graph is built directly from high-dimensional observations (e.g., images). Each state is a vertex, and two vertices are connected with an edge if the agent transitions between them during training time. As a result, trajectories are joined when a common state appears in two different rollouts. In our framework, graph building and exploration proceed iteratively. The resulting graph provides information about the predecessor of each state, which is used to determine the update order of the $Q$-function based on the reverse sweep principle. The reverse sweep is initiated from a set of terminal states because bootstrapping is not required to determine the correct $Q$-value for these states. We call this method *Topological Experience Replay* (TER) because the update order of $Q$-values is based on the topology of the state space.

## 2 RELATED WORKS

**Experience replay**   Devising an experience replay strategy to improve $Q$-learning is an active field in deep RL. *Prioritized Experience Replay (PER)* (Schaul et al., 2015) prioritize states by the bootstrapping errors (also called as temporal difference (TD) error) (Schaul et al., 2015). Since during training the TD error depends on a (possibly) erroneous estimate of the $Q$-values, PER can lead to slow convergence speed. *Episodic Backward Update (EBU)* (Lee et al., 2019) and Lin (1992) replays interactions in trajectories in the reverse order of state visitations in every episode. However, for instance, in Figure 1, moving backward in a trajectory $(A, E, B, D, G)$ misses the other predecessors of $D$ (i.e., $C$) while the $Q$-value of $D$ affects that of $C$. *DisCor* (Kumar et al., 2020) re-weights the error of state transitions to approximate the optimal bootstrapping order while the conservative weighting scheme of DisCor impedes learning progress. Different from the above works, we reorder states for updates based on the states' $Q$-value dependency.

**Reverse sweep**   Similar ideas of reverse sweep were studied in different scenarios. Assuming the access to the simulators, Florensa et al. (2017) trained RL agents from initial states that are closer to the goal and are gradually chosen to be farther away. Moore & Atkeson (1993) fitted an environment dynamics model for generating the predecessors of the states of high bootstrapping errors successively backward to update the Q-values. Goyal et al. (2019) extended this algorithm by a learned generative model to perform backward rollouts from high-value states. Schroecker

et al. (2019) trained an imitation learning model to match the expert's predecessor state distribution generated by a backtracking model. Instead of augmenting the training dataset by manipulating simulators or generating synthetic data, our method rearranges the existing dataset to aid training.

**Graph for RL** Prior works organizing experience into a graph focused on temporally abstracted planning. Savinov et al. (2018) learn to build a graph for navigation from images. Eysenbach et al. (2019); Zhang et al. (2021); Emmons et al. (2020) provide RL agents with sub-goals by planning over graph. Zhu et al. (2020) regularizes the $Q$-learning training via an additional graph $Q$-function defined over graph. Instead of taking advantages of temporal abstraction from graph, we use a graph to maintain states' dependency for improving $Q$-function learning.

**Model-based RL** Graph is similar with the world model used in model-based RL, but TER uses it for a different purpose. The classic work (Sutton, 1991) maintains the state transition probabilities of each state and sample states by these transition probabilities to update the $Q$-values. Recent model-based methods (Hafner et al., 2019; Nagabandi et al., 2018; Deisenroth & Rasmussen, 2011) typically learn a world model (e.g., neural network or Gaussian process) for generating "imaginary" data for training the RL agent or planning. Imaginary data augments the replay buffer, improving the agent's performance by training it with data absent in the replay buffer. Instead of generating synthetic state transitions, TER employs the graph to find a better update order of each state transition in the existing replay buffer for training the value function. In other words, model-based RL augments the dataset while TER sorts the dataset.

## 3 PRELIMINARIES

We consider reinforcement learning (RL) (Sutton & Barto, 2018) in an episodic discrete-time Markov decision process (MDP). The objective of RL is to find the optimal policy, $\pi^*$, that maximizes expected return $\mathbb{E}\left[\sum_{\tau=t}^{T-1} \gamma^{\tau-t} r_\tau | s_t = s\right] \forall s \in \mathcal{S}$, where $\gamma$ is a discount factor (Sutton & Barto, 2018), the $\mathcal{S}$ represents the set of all states in the MDP, and $\tau$ denotes the time step. At a time step $\tau$, the agent takes the action $a_\tau = \pi(s_\tau)$, receives reward $r_\tau = (s_\tau, a_\tau, s_{\tau+1})$, and transitions to the next state $s_{\tau+1}$, where $R$ is the reward function. At the time $\tau$ of task completion (e.g., reaching goal states) the episode termination indicator $\mathcal{E}(s_\tau) = 1$, otherwise $\mathcal{E}(s_\tau) = 0$.

$Q$-learning is a popular algorithm for finding the optimal policy. It learns the $Q$ function $Q(s, a) = \mathbb{E}\left[\sum_{\tau=t}^{T-1} \gamma^{\tau-t} r_\tau | s_t = s, a_t = a\right]$ using bootstrapping operations: $Q(s, a) \leftarrow \mathcal{R}(s, a, s') + \gamma \max_{a'} Q(s', a')$ where $s'$ is the state encountered on executing action $a$ in the state $s$. The policy $\pi(s)$ can be easily derived as: $\pi(s) := \arg\max_a Q(s, a)$. When $\mathcal{S}$ is high-dimensional, the $Q$ function is usually represented by a deep neural network (Mnih et al., 2015). The interaction data collected by the agent is stored in an experience replay buffer (Lin, 1992) in the form of state transitions $(s_t, a_t, r_t, s_{t+1})$. The $Q$ function is updated using stochastic gradient descent on batches of data randomly sampled from the replay buffer.

## 4 METHOD

Our method, topological experience replay (TER), is central at performing *reverse sweep* to update the $Q$-function in high-dimensional state spaces. Reverse sweep is known to accelerate $Q$-function convergence yet requires the knowledge of predecessors of a state, which is often unknown in high-dimensional state spaces. We overcome this limitation by building a graph from the replay buffer and guide the $Q$-function update with a reverse sweep over the graph. Section 4.1 presents the overview of our proposed algorithm. In Section 4.2, we describe the procedure for building the graph from high-dimensional states. Next, in Section 4.3, we describe how the graph is used to determine the update order for $Q$-learning. Finally, we present a batch mixing technique in Section 4.4 that guarantees TER to converge.

### 4.1 TOPOLOGICAL EXPERIENCE REPLAY

**Assumption** Similar to prior works of reverse sweep in tabular domains (Dai & Hansen, 2007), we assume that the objective of the MDPs is to reach terminal states (i.e., goal states). When an

agent successfully completes the task, the agent must be at some terminal states. This assumption is required, otherwise reverse sweep will not help the agent learn optimal $Q$-value faster. Another assumption is that an agent must be able to visit the same state twice so that we can find the joint state between two episodes for building the graph. This is an admissible assumption since Zhu et al. (2020) shows that an agent can visit repeated states even in high-dimensional image state spaces Bellemare et al. (2013).

**Overview**   TER iteratively interleaves between *graph building* and *reverse sweep* online during training. The agent collects experience at each step and periodically updates the $Q$-function. Each new transition $(s, a, r, s')$ is mapped to an edge on the graph. When updating the $Q$-function, TER performs a few steps of reverse sweep for making a batch of training samples. The reverse sweep starts from the vertices associated with terminal states. Each step in the sweep gathers transitions stored at an edge for training the $Q$-function. The batch mixing technique mixes the transitions that are fetched during reverse sweep and randomly sampled from the replay buffer, ensuring those transitions disconnected from terminal states to be updated. The pseudo-code in Algorithm 1 details the whole procedure.

## 4.2   GRAPH BUILDING

**Data structure**   We organize the replay buffer as an unweighted graph of state transitions. The graph keeps track of the predecessors of states and stores the transitions. Let an unweighted graph be $\mathcal{G} = \{V, E\}$, where $V, E$ denote the set of vertices and edges, respectively. As the graph will be queried frequently for replaying experience, we desire a query-efficient data structure. Thus, we implement $\mathcal{G}$ as a nested hash table, where the outer ($v'$) and inner ($v$) keys correspond to a pair of vertices in an edge $e(v, v') \in E$ where $v, v' \in V$. We associate a transition $(s, a, r, s')$ with an edge by a hash function $\phi$. Each edge $e(v, v')$ stores a list of transitions $(s, a, r, s')$ where $v = \phi(s)$ and $v' = \phi(s')$. We then can retrieve the predecessor transitions of a state in $O(1)$ time.

**Hashing**   Since high-dimensional states cannot directly serve as the hash-tables keys for fast query, we require a encoding function $\phi : \mathcal{S} \mapsto \mathbb{R}^d$ that transforms states to low-dimensional representations (keys in the graph $\mathcal{G}$). The minimal requirements of this encoding function is being able to distinguish two different states. We thus make use of random projection as the encoding function $\phi(s) := \mathbf{M}s$ where $s \in \mathcal{S}$, $\mathbf{M}$ is a matrix with elements randomly sampled from a normal distribution `Normal`$(0, 1/d)$, where $d$ denotes the dimension of projected vectors[1], as prior work has empirically found that random projections can distinguish individual visual observations (Burda et al., 2019; Zhu et al., 2020). Though an encoding function that aggregates states with similar task-relevant features would be helpful for determining better update orders, we simply use random projection, since our focus is not representation learning for hashing.

**Update scheme**   The graph is iteratively updated online at each time step by the incoming transition and each transition $(s_t, a_t, r_t, s_t)$ is stored into the hash table $\mathcal{G}$. We add $\phi(s_t)$ and $\phi(s_{t+1})$ to the vertex set $V$ and append the transition $(s_t, a_t, r_t, s_{t+1})$ to the list of transitions stored in the edge $e(\phi(s_t), \phi(s_{t+1}))$. We can retrieve the predecessor transitions of $s'$ by querying $e(v, \phi(s'))\ \forall v \in V$ for a reverse sweep. The predecessor transitions of a state is aggregated from multiple trajectories. Let $(s_A, a_A, r_A, s'_A)$ and $(s_B, a_B, r_B, s'_B)$ be two transitions from different trajectories. Both transitions can be retrieved by $e(v, v')\ \forall v \in V$ if $v' = \phi(s'_A) = \phi(s'_B)$ (i.e., outer keys of $\mathcal{G}$).

## 4.3   REVERSE SWEEP

**Algorithm**   Once the graph is built, the next step is to determine the ordering of states for updating the $Q$-values. Same as common experience replay methods, $Q$-function is periodically updated online with mini-batches of data sampled using a fixed replay ratio (Fedus et al., 2020). Each batch of training data is collected via a reverse sweep. More specifically, we maintain a record of vertices that correspond to terminal states, and this set is denoted by $V_\mathcal{E}$. Each terminal vertex $v_e$ acts as the root node. We perform a reverse breadth-first search (BFS) on the graph starting from a subset of $v_e$ sampled from $V_\mathcal{E}$ separately. The $v_e$ serves as the initial frontier vertex $v'$. For a frontier vertex $v'$

---

[1]If $s$ is an image, we flatten $s$ into an one dimensional vector before the transformation.

in the search tree, we first sample its predecessors $v$ and append the corresponding state-transitions sampled from $e(v, v')$ to a `batch_queue`. Then we set all the predecessors $v$ to be the new frontier vertices and fetch transitions. Once there are $B$ transitions in the `batch_queue`, we pop out $B$ state-transitions for updating the $Q$-function. We resume the tree search next time starting from the frontier $v'$ to update the $Q$-function. Note that BFS only expands each vertex once, and thus infinite loop on cyclic (i.e., bi-directional) edges will not happen. When there is no vertex to expand, reverse sweep is restarted from a set of root vertices sampled from $V_\mathcal{E}$ again.

**Complexity** Compared with uniform experience replay (UER) (Lin, 1992; Mnih et al., 2015), the extra memory needed is at most $|E|$ low-dimensional vectors in the hash-table's keys. Instead of growing endlessly, the graph $\mathcal{G}$ is pruned once the number of transitions $(s, a, r, s')$ on the graph is greater than a user-specified replay buffer capacity. For pruning details and the hyperparameter settings, see the Section A. The time complexity of making a batch of training transitions is $O(B)$ as we fetch data from $B$ vertices. We show that the wallclock computation time is close to typical methods in Section A.7.

## 4.4 BATCH MIXING

One potential drawback of starting the reverse sweep from only terminal states is that the states that are unreachable from the terminal states will never be updated. Consequently, the $Q$-values of such states are not updated, which impedes the convergence of $Q$-learning. Prior work (Dai & Hansen, 2007) has shown that interleaving value updates at randomly sampled transitions and at transitions selected by any prioritization mechanism ensure the convergence. Thus, we mix experience from TER and PER Schaul et al. (2016) to form training batches using a mixing ratio $\eta \in [0, 1]$. For each batch, $\eta$ fraction of the data is from PER, and $1 - \eta$ is from TER. Additional details of batch mixing are provided in the supplementary material Section A.

---

**Algorithm 1** Topological Experience Replay for Q-Learning
> **Input:** A hash function $\phi$, a warm up period $T_\text{warm}$, a batch size $B$

1: Set empty graph $\mathcal{G} \leftarrow \{V = \emptyset, E = \emptyset\}$ and terminal vertices set $V_\mathcal{E} \leftarrow \emptyset$
2: Set `search_queue` and `batch_queue` as empty queues and `visited_vertex` $\leftarrow \emptyset$ for BFS
3: **for** $t = 0, 1, \dots$ **do**                                          $\triangleright$ online training loop
4:     Take action $a_t$ and receive experience $(s_t, a_t, r_t, s_{t+1})$
5:     Add $\phi(s_t)$ and $\phi(s_{t+1})$ to $V$, and $e(\phi(s_t), \phi(s_{t+1}))$ to $E$
6:     Augment the terminal vertices set $V_\mathcal{E} \leftarrow V_\mathcal{E} \cup \{\phi(s_{t+1})\}$ if $\mathcal{E}(s_{t+1}) = 1$
7:     **while** $t > T_\text{warmup}$ and `len(batch_queue)` $< B$ **do**
8:         **if** `search_queue` is empty **then**
9:             Add sampled vertices from $V_\mathcal{E}$ to `search_queue`, `visited_vertex` $\leftarrow \emptyset$
10:         **end if**
11:         Pop $v'$ from `search_queue` until $v' \notin$ `visited_vertex`
12:         Sample a subset of the predecessors edges $E_P(v') = \{e(u, u') \in E \mid u' = v'\}$ of $v'$
13:         Push $v \in \{v \mid e(v, v') \in E_P(v')\}$ to `search_queue`
14:         Push $(s, a, r, s')$ stored at each predecessor edge $e(v, v') \in E_P(v')$ to `batch_queue`
15:         Mark $v'$ as expanded: `visited_vertex` $\leftarrow$ `visited_vertex` $\cup \{v'\}$
16:     **end while**
17:     **if** $t > T_\text{warmup}$ **then**
18:         Pop a minibatch of experience $b = \{(s_i, a_i, r_i, s_i')\}_{i=1}^B$ from `batch_queue`
19:         Update $Q$ function with $b$
20:     **end if**
21: **end for**

---

## 5 EXPERIMENTS

We aim to answer the following primary questions: (1) Does TER accelerate the speed of convergence of $Q$-learning? (2) Does TER improve sample efficiency in high-dimensional state spaces?

(3) Does baselines with higher replay ratios match TER's efficiency? In addition, Section 5.5 analyzes the effect of batch mixing in TER, and Section 5.6 investigates why TER improves the sample efficiency.

## 5.1 SETUP

**Environments** We evaluate TER in *Minigrid* (Chevalier-Boisvert et al., 2018) and *Sokoban* (Schrader, 2018) as the tasks' objectives are to reach terminal states (i.e., goal states) and both domains have challenging tasks for $Q$-learning. In *Minigrid* and *Sokoban*, the agent observes the top-down image view of a grid map and moves around in the grid map to complete the task. For each task, the map is randomly generated at each episode, so the policy needs to generalize across different map configurations. In *Sokoban*, the task is to push all the boxes on the map to the target positions where the map sizes and the number of boxes can be varied. We denote a configuration in the format `<size>x<size>-<num. boxes>`. All of the environment details can be found in the Section C.

**Baselines.** We compare TER with uniform experience replay (UER) (Lin, 1992; Mnih et al., 2015), prioritized experience replay (PER) (Schaul et al., 2016), episodic backward update (EBU) (Lee et al., 2019), and DisCor (Kumar et al., 2020). UER uniformly samples state-transition tuples from the replay buffer. PER prioritizes experience with high temporal difference (TD) (Sutton, 1988) errors defined as $|r_t + \max_{a'} Q(s', a') - Q(s, a)|$. EBU uniformly samples an episode from the replay buffer, and updates the $Q$-values of the state-action pairs in the sampled episode reversely. For example, given an episode $[s_1, a_1, r_1, s_2, a_2, r_2, s_3, \ldots s_{T-1}, a_{T-1}, r_{T-1}, s_T]$, EBU updates in a sequence $[Q(s_{T-1}, a_{T-1}), Q(s_{T-2}, a_{T-2}), \ldots, Q(s_1, a_1)]$. DisCor re-weights each $Q$-function update inversely proportional to the estimated bootstrapping error $|Q^*(s', a') - Q(s', a')|$ of a state-action pair $(s, a)$, where $Q^*$ denotes the optimal $Q$-function.

**Implementation.** We test each experience replay method along with double deep Q-learning (DQN) (Van Hasselt et al., 2016) with the same architecture used by (Mnih et al., 2015). In DQN, the $Q$ function is updated once per four environment steps (i.e., replay ratio $= 0.25$) unless specified otherwise. We use $\epsilon$-greedy method for exploration. We have reused implementations of the baseline methods and DQN from `pfrl` codebase (Fujita et al., 2021). For EBU, we adapted the publicly released code [2] in the framework of `pfrl`. We re-implemented DisCor and verified that our implementation reproduces the results reported in the paper (Kumar et al., 2020). For TER, we use $\eta = 0.5$ for *Minigrid* and $\eta = 0.2$ for *Sokoban*. Additional implementation details and our reproduction of DisCor results are available in Section A.

**Evaluation Metric.** We ran each experiment with 5 different random seeds and reported the mean (solid or dashed line) and $95\%$-confidence interval (computed by bootstrapping method DiCiccio et al. (1996)) on the learning curves. The learning curves indicate the average normalized return over 100 testing episodes in a varying number of environment steps. More details are provided in Section B.

## 5.2 AN ILLUSTRATIVE EXAMPLE

We experimented with *NChain* environment, as shown in Figure 2a, to illustrate that TER gives rise to a more efficient updating order. The only positive reward in *NChain* emits at the rightmost terminal state (i.e., node $N$). As the only reward is distant from the starting state, it is easy to observe the impact of a misspecified update order. In order to rule out the influence of exploration, we collected a fixed number of rollout episodes where the agent took random actions and then used these pre-collected experiences to train TER and all the baselines. Figure 2b shows the normalized return and the average $Q$-value error as the training progresses. The value error is calculated by the absolute difference between the agent's $Q$-function and the ground truth optimal $Q$-function. We see that the TER agent obtains the highest score after only 30 value backups, while UER, PER, and DisCor fail to solve the task in 100 updates, which shows the importance of update order. Even the best baseline method (EBU) takes twice the number of backups on average. As TER differs

---

[2]https://github.com/suyoung-lee/Episodic-Backward-Update

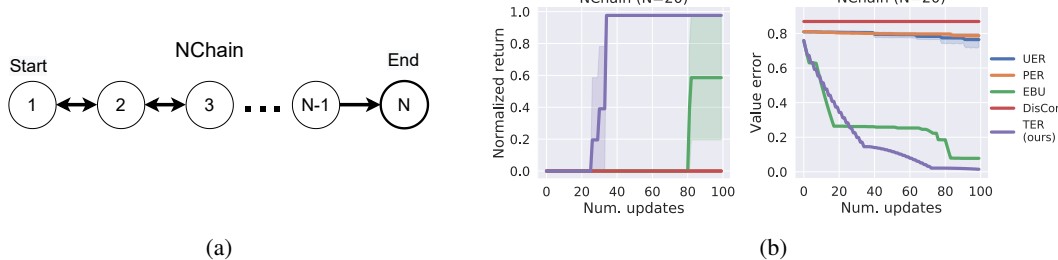

(a)                                                                  (b)

Figure 2: **(a)** The illustration of *NChain* environment. Each node denotes a state, and the arrow represents a valid transition between two nodes. **(b)** TER achieves higher performance with fewer updates and reduces the value error of the $Q$-function faster than the baselines.

from EBU in that EBU only backtracks from the instantaneous trajectories, this shows the benefits of backtracking from the episodes stitched by the graph. Also, we show that the value error of TER reduces faster than the baselines, which suggests that TER converges to the ground truth optimal $Q$-function faster.

### 5.3 HOW DOES TER PERFORM IN A HIGH-DIMENSIONAL STATE SPACE?

We investigated if TER is effective in environments with a high-dimensional state space such as images in *Minigrid* and *Sokoban* (see Section 5.1). Overall, TER outperforms the baselines in *Minigrid* and *Sokoban*, as shown in Figure 3a and Figure 3b. In *Sokoban*, such improvement is even more significant when the environment map is bigger, and there are more boxes than the others. Surprisingly, we can see from Figure 3b that EBU and DisCor perform worse than UER. We find that the average $Q$-value of EBU is much bigger than the maximum possible return in the environment (Section D), suggesting that EBU suffers from the overestimation problem (Hasselt, 2010) that hinders the $Q$-learning. Also, we find that the exploding value error estimates in DisCor impede the learning progress, where the updates are down-weighted exceedingly due to huge estimates on value error. In addition, we present the analysis on the influence of random projection dimension $d$ in Section J, showing that TER is insensitive to the choice of $d$.

### 5.4 HOW DOES TER COMPARE TO THE BASELINES WITH A HIGHER REPLAY RATIO?

As we mentioned in Section 1, the convergence speed of $Q$-learning can affect sample efficiency. Even though TER demonstrates superior learning efficiency in the previous section, a critical unanswered question is whether baselines can achieve similar sample efficiency with more frequent $Q$-function updates compared to TER. Thus, for all baselines, we increase the replay ratio defined as the number of gradient updates per environment transition (Fedus et al., 2020). Figure 4 shows the learning curves of the baselines with different replay ratios of $0.25, 1.0, 2.0$ and TER. For instance, $0.25$ means 1 gradient update for every 4 environment steps. Overall, increasing the replay ratio does not consistently improve the performance of the baselines, except for *LavaCrossing-Hard*. We found that a possible reason could be $Q$-value overestimation (Hasselt, 2010). Figure 4b shows that the average $Q$ values of the baselines with larger replay ratios are much higher than $R_{\max}$ (i.e., the maximum return in the task), implying a significant $Q$-value over-estimation. We compute the normalized difference between the learned $Q$ values and $R_{\max}$: $\frac{\mathbb{E}_{s,a}[Q(s,a)] - R_{\max}}{R_{\max}}$, and $\mathbb{E}_{s,a}[Q(s,a)]$ is the average $Q$ values during training. Therefore, even though we increase the replay ratio, it does not close the performance gap between the baselines and TER.

### 5.5 HOW DOES BATCH MIXING AFFECT TER?

Figure 5 shows the learning curves of TER with different batch mixing ratios, where the number in the legend denotes the batch mixing ratio $\eta$ defined in Section 4.4. The results of *Sokoban* show that a high ratio can fail TER and a low ratio leads to premature convergence. It can be seen that the best-performing variants are TER with $0.1$ and $0.2$ mixing ratios. On the contrary, TER is insensitive to the mixing ratio in *Minigrid*, where each variant exhibits a similar performance. The reason for

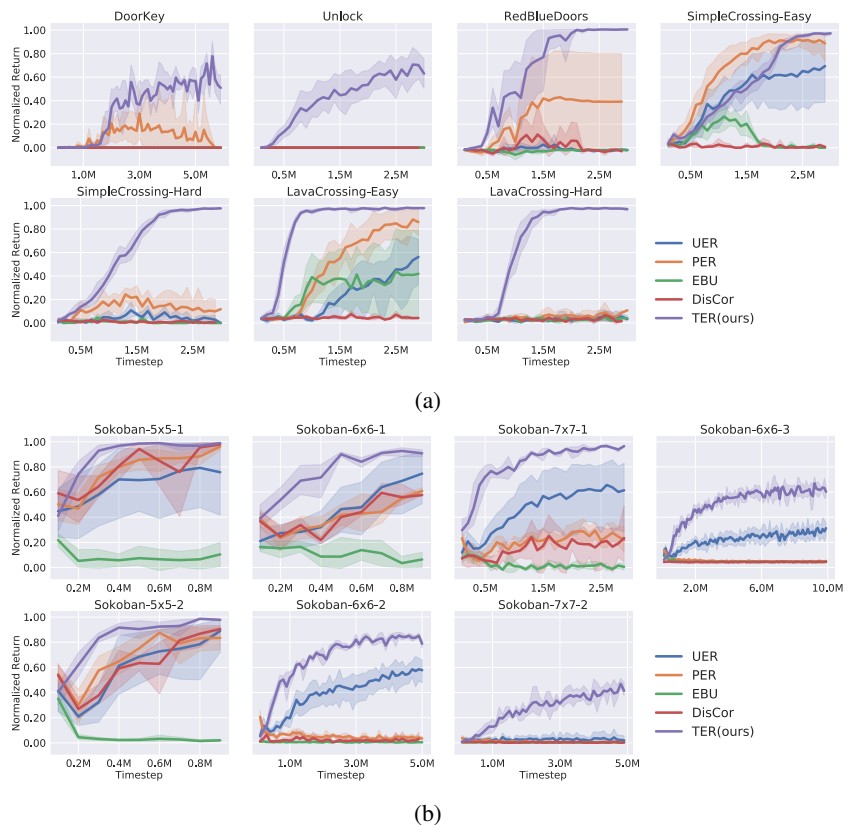

Figure 3: TER outperforms the baselines in (a) *Minigrid* and (b) *Sokoban*, demonstrating that TER can work with high-dimensional state spaces. In (b), the plots are ordered by task difficulty (i.e., reward sparsity) from left to right. The performance gain of TER is more salient when rewards are more sparse. The $x$-axis is the environment steps. $y$-axis is the normalized mean return.

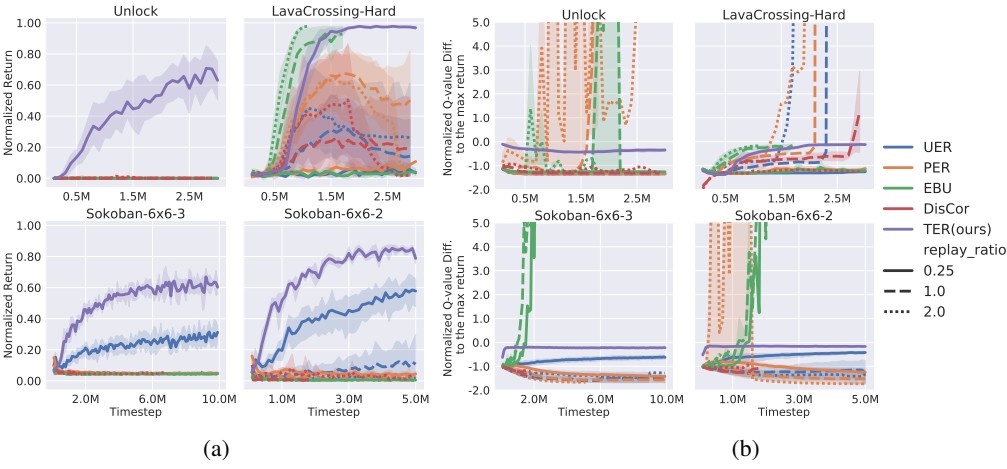

Figure 4: **(a)** Increasing the replay ratio does not consistently improve the performance of the baselines. **(b)** Increasing replay ratios leads to $Q$-value overestimation in baselines.

this result could be that the successful experience is easier to get in *Minigrid* than that in *Sokoban*. As the states in a successful episode must be connected to a terminal state, more transitions will be replayed in the reverse sweep. Hence the issues about the non-updated states can be minor. Overall, though the choice of mixing ratio can be environment dependent, we empirically found that TER with mixing ratio 0.1 and 0.2 perform well in both *Minigrid* and *Sokoban*.

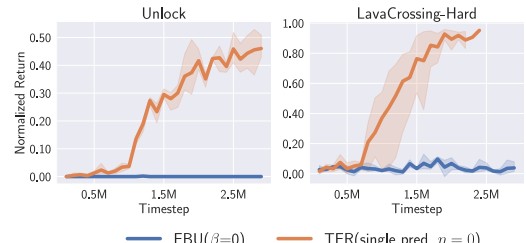

Figure 5: The learning curves of TER with different batch mixing ratios. In *Sokoban* the choice of mixing ratio is more sensitive than that in *Minigrid*. TER with mixing ratio $0.1$ and $0.2$ perform well in both *Minigrid* and *Sokoban*.

### 5.6 WHY DOES TER OUTPERFORM EBU?

Since EBU and TER share the idea of reverse replay, a natural question is what explains the significant performance gap between the algorithms reported in Section 5.3. To investigate if the benefit originates from stitching episodes, we made a variant of TER that disables batch mixing (i.e., $\eta = 0$) and uses only one search tree and sample one predecessor during BFS, called *TER(single pred, $\eta = 0$)*. Also, to rule out the influence of $Q$-value mixing hyperparameter $\beta$ (Lee et al., 2019) and timeout-states in EBU, we set $\beta = 0$ for EBU and only replay backward from the terminal states in this experiment[3]. As such, the only difference between *TER(single pred, $\beta = 0$)* and EBU is the ability to replay backward from a new path through the joint state between different trajectories. Figure 6 shows that *TER(single pred, $\eta = 0$)* outperforms EBU, suggesting that the performance gap between TER and EBU likely results from trajectory stitching. In addition to this primary finding, we also show that applying batch mixing and filtering out timeout-states for EBU do not improve EBU's performance in Section H and Section I.

Figure 6: The only difference between *TER(single pred, $\eta = 0$)* and *EBU ($\beta = 0$)* is trajectory stitching while *TER(single pred, $\eta = 0$)* still outperforms *EBU ($\beta = 0$)*. This shows that trajectory stitching results is critical for TER's performance gain.

## 6 CONCLUSION

In conclusion, we showcased that replaying experience in a backward topological order expedites $Q$-learning in goal-reaching tasks. Moreover, our experiments demonstrated that TER works in cyclical MDPs even though the strict topological orders are unclear where the rationale is presented in Section M. We present more discussion in Section O.

### ACKNOWLEDGEMENT

We thank members of Improbable AI Lab for the helpful discussion and feedback. We are grateful to MIT Supercloud and the Lincoln Laboratory Supercomputing Center for providing HPC resources. The research in this paper was supported by the MIT-IBM Watson AI Lab and it part by the Army Research Office and was accomplished under Grant Number W911NF-21-1-0328. The research was also supported by the Army Research Office and was accomplished under Grant Number W911NF-21-1-0328. The views and conclusions contained in this document are those of the authors and should not be interpreted as representing the official policies, either expressed or implied, of the Army Research Office or the United States Air Force or the U.S. Government. The U.S. Government is authorized to reproduce and distribute reprints for Government purposes notwithstanding any copyright notation herein.

---

[3]Timeout-state does not indicate the task completion but triggers resets due to running out of time. EBU also replays backward from these states.

## 7    CODE OF ETHICS

Not applicable.

## 8    REPRODUCIBILITY STATEMENT

Please refer to Section A.

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

# A APPENDIX

# A IMPLEMENTATION DETAIL

We implement all the methods based on (Fujita et al., 2021).

## A.1 DEEP Q-NETWORK IMPLEMENTATION DETAIL

We use double DQN (Van Hasselt et al., 2016) for all the methods.

**Network architecture** Our networks consist of two types of layers: 2-D convolutional layer `Conv2D` and fully connected layer `Linear`. We denote a `Conv2D` layer with parameters (`input_channel_dim, output_channel_dim, kernel_size, stride`) and `Linear` with (`input_channel_dim, output_channel_dim`).

For *Minigrid*, the DQN architecture is the sequential combination of [`Conv2D(40*40*3, 32, 8, 4), Conv2D(32, 64, 4, 2), Conv2D(64, 64, 3, 1), Linear(64, 512), Linear(512, num_actions)`] with `ReLU` activation units followed by each layer.

For *Sokoban* and *Atari*, the DQN architecture is [`Conv2D(84*84*3, 32, 8, 4), Conv2D(32, 64, 4, 2), Conv2D(64, 64, 3, 1), Linear(3136, 512), Linear(512, num_actions)`] with `ReLU` activation units.

**Target network** For all environments, we use `hard-reset` that copies the current $Q$-function network to the target network periodically. The target network update interval is 1000 for *Minigrid* and $10,000$ for *Sokoban* and *Atari*.

**Batch size, Optimizer, and Learning rate** For all environments, we set `batch_size=64` for *Minigrid*, `batch_size=32` for *Sokoban* and *Atari*. The optimizers are `Adam` with `learning_rate=3e-4` for *Minigrid* and *Sokoban*. For *Atari*, we follow the configuration in (Mnih et al., 2015) and use `RMSProp` optimizer with `learning_rate=2.5e-4, alpha=0.95, eps=1e-2`, and `momentum=0.0`.

**$\epsilon$-greedy exploration** We use $\epsilon$-greedy exploration for our method and the baselines. The value of $\epsilon$ is linearly decayed from $1.0$ to $0.01$ in $1M$ environment steps.

**Experience Replay buffer** We warm-up the replay buffer with the trajectories where the agent takes random actions. The warm-up periods are $50,000$ for all environments. The capacity (i.e., maximum number of elements) of the buffer is $1M$ for each environment. We draw 1 batch of experience from the replay buffer to update the $Q$-function every 4 environment steps, except for the experiments in Section 5.4.

## A.2 PRIORITIZED EXPERIENCE REPLAY (PER) (SCHAUL ET AL., 2016) IMPLEMENTATION DETAILS

We use the implementation provided in `pfrl`. There are two hyper-parameters in PER: $\alpha$ and $\beta$. Following the original paper (Schaul et al., 2016), we set $\alpha = 0.6$ and $\beta = 0.4$. The $\beta$ linearly transitions to $1.0$ within $\frac{1}{4}$ of the total training environment steps in the environment.

## A.3 EPISODIC BACKWARD UPDATE (EBU) (LEE ET AL., 2019) IMPLEMENTATION DETAILS

We adapted the code from the official code-base [1] to `pfrl` and set diffusion factor (Lee et al., 2019) as $0.5$. Note that we also use double DQN for EBU, though the original paper (Lee et al., 2019) uses the vanilla DQN (Mnih et al., 2015). As a sanity check for the correctness of implementation, we ran EBU in some of the *Atari* environments reported in the original paper (Lee et al., 2019). The results shown in Figure L.15 show that the performances of our reproduced results are higher than

---

[1] https://github.com/suyoung-lee/Episodic-Backward-Update

|        | Mean (sec.) | Median (sec.) | Min (sec.) | Max (sec.) |
|--------|-------------|---------------|------------|------------|
| UER    | 0.002       | 3e-5          | 1e-5       | 0.086      |
| PER    | 0.003       | 0.0001        | 4e-5       | 0.28       |
| EBU    | 0.002       | 3e-5          | 1e-5       | 0.083      |
| DisCor | 0.003       | 4e-5          | 1e-5       | 0.11       |
| TER    | 0.002       | 0.0002        | 0.0001     | 0.256      |

Table A.1: Computation time of making a batch of training transitions over the training time.

or match the original results in *Pong*, *Kangaroo*, and *Venture*, but lower than the original ones in *Freeway*.

## A.4 DISCOR (KUMAR ET AL., 2020) IMPLEMENTATION DETAILS

We re-implemented DisCor based on the implementation details provided in the original paper (Kumar et al., 2020). The initial temperature $\tau$ is set as 10.0 and the $\tau$ is updated to the batch mean of error network $\Delta_\phi$ prediction over the last target update interval when the target network is updated. As the original paper suggests, the architecture of the error network $\Delta_\phi$ is the DQN architecture for each task with an additional fully connected layer at the end.

We verified the correctness of our implementation by testing DisCor in Atari Bellemare et al. (2013) *Pong* that is reported in the original paper (Kumar et al., 2020). Figure L.15 shows our result of *Pong* matches the original result in (Kumar et al., 2020).

## A.5 TER IMPLEMENTATION DETAILS

**Reverse Breadth-first Search (BFS)**   Since the number of terminal states in the environment can be large, we sample the vertices of some terminal states as the roots of BFS search trees. For all environments, we sample 8 terminal vertices as the roots. For each vertex, we sample at most 3 predecessors to expand instead of enumeration. Enumerating all the predecessors can have too many predecessors

**Vertex Encoding**   We set the dimension $d$ of the encoded states as 3 for all the environments.

**Batch Mixing**   We set the batch mixing ratio $\eta$ as 0.5 for *Minigrid* and 0.1 for *Sokoban*.

**Graph Pruning**   We prune the graph when the number of transitions stored at the edges in the graph exceeds the replay buffer capacity. The pruning is carried out according to the "age" of each transition. The age of a transition is `current_env_step - transition_env_step`, where `current_env_step` is the current environment steps and `transition_env_step` the environment step when it was added to the buffer. When sampling a batch of experience from the `batch_queue` in Algorithm 1, we remove the transition of which age exceeds the replay buffer capacity from the current batch and from the corresponding edge and pop more transitions from `batch_queue` to replace the removed transitions. Once an edge has no transitions remaining, we remove the edge from the graph.

## A.6 CODE

Code is included in the zip file.

## A.7 COMPARISON OF COMPUTATION TIME

As TER requires performing graph search over the training time, it is worth knowing how much extra computation time is required by TER. Table A.1 shows that TER's running time is similar to canonical experience replay methods on average. Also, the maximum running time over the training process is not greater than that of PER.

## B    EVALUATION DETAIL

We ran each experiment with 5 different random seeds and report the mean (solid or dashed line) and bootstrapped confidence interval (shaded part) (DiCiccio et al., 1996) on the learning curves. The learning curves show how the averaged normalized return over 100 testing episodes change over the number of environment interaction steps.

For each testing episode, we fix the neural network weights of $Q$-function and run the agent with the greedy action (i.e., $\arg\max_a Q(s, a)$). Since the greedy action will not change if the state remains the same, we let the agent take random actions with probability $0.05$ to prevent the agent from being stuck at the same state.

## C    ENVIRONMENT DETAIL

We illustrate the example maps of each environment in Figure C.1.

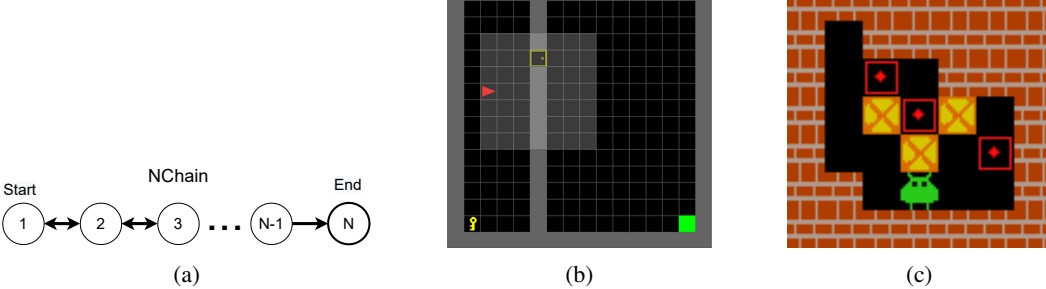

(a)                              (b)                              (c)

Figure C.1: **(a)** *NChain.* Each circle denotes a state and arrow represents a valid transition between two circles. **(b)** *Minigrid.* This figure shows one example environment in *Minigrid*. The agent takes the top-down image view of the two dimensional grid as states. The red arrow denotes the agent, the green cell is the goal, the yellow square is a door, and the yellow key is used for opening the door. The black and the gray cells are walk-able areas and walls respectively. The task is to pickup the key, open the door, and then go to the goal. **(c)** *Sokoban.* A randomly sampled environment in *Sokoban* domain. The yellow block denotes a box, the red square represents a target position, and the green object is the agent. The outer brown bricks denote walls and the black cells are walk-able ares. The goal is to push all the boxes to the target positions.

### C.1    CHAIN

For *NChain* shown in Figure C.1a, there are $N$ states, $s_1 \ldots s_N$ in the state space $\mathcal{S}$ and $\{\texttt{forward}, \texttt{backward}\}$ actions in the action space $\mathcal{A}$. At each state $s_i$, the agent can move to the next state $s_{i+1}$ by $\texttt{forward}$ and to the last state $s_{i-1}$ by $\texttt{backward}$. If the agent is at $s_1$, the action $\texttt{backward}$ will not effect. The agent gets $+1$ reward when transitioning from $s_{N-1}$ to $s_N$ and $0$ otherwise. An episode terminates when the agent is at $s_N$.

### C.2    MINIGRID

The official *Minigrid* implementation is referred to (Chevalier-Boisvert et al., 2018). Figure C.1 shows an example top-down view image of the map. For each environment, each state is an RGB top-down view image resized to to $40 \times 40$ and the action space is $\{\texttt{turn\_left}, \texttt{turn\_right}, \texttt{forward}, \texttt{pickup}, \texttt{drop}, \texttt{toggle}\}$. An episode terminates when the timestep exceeds the maximum time steps MAXSTEPS dependent on the task or the agent receives +MAXSTEPS reward. Note that we modify the reward function defined in (Chevalier-Boisvert et al., 2018) since the default reward function is history dependent function. A history-dependent reward function is not valid for fully observable MDPs since history data is un-observable in the state.

**DoorKey**  The example levels are illustrated in Figure C.2. The task is to pickup the key (yellow key icon), open the door (yellow square icon), and navigate to the goal (green block). The reward is defined as

$$\mathcal{R}(s, a, s') = \begin{cases} +\texttt{MAXSTEPS}, & \text{the goal is reached} \\ -1, & \text{otherwise} \end{cases}$$

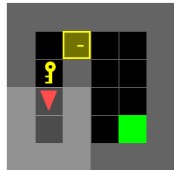 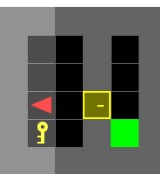 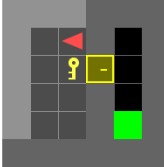 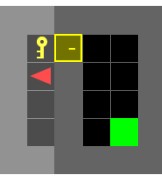 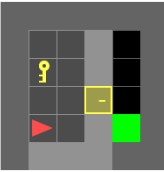

Figure C.2: The images of randomly generated levels of *DoorKey*

**Unlock**  The example levels are illustrated in Figure C.3. The task is to pickup the key (key icon) and open the door (yellow square icon). The reward is defined as

$$\mathcal{R}(s, a, s') = \begin{cases} +\texttt{MAXSTEPS}, & \text{the door is opened} \\ -1, & \text{otherwise} \end{cases}$$

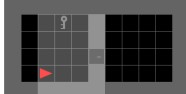 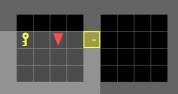 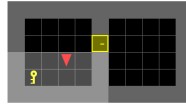 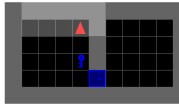 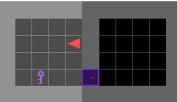

Figure C.3: The images of randomly generated levels of *Unlock*

**RedBlueDoors**  The example levels are illustrated in Figure C.4.  he task is to open the red door (red square) and open the blue door (blue door). The reward is defined as

$$\mathcal{R}(s, a, s') = \begin{cases} +\texttt{MAXSTEPS}, & \text{the red door and the blue door are opened sequentially} \\ -1, & \text{otherwise} \end{cases}$$

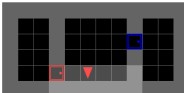 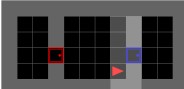 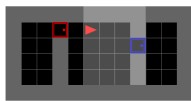 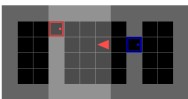 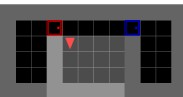

Figure C.4: The images of randomly generated levels of *RedBlueDoor*

**SimpleCrossing-Easy/Hard**  The example levels are illustrated in Figure C.5a and Figure C.5b. The task is to navigate through the rooms to reach the goal (green block). There are more rooms in the hard one.

$$\mathcal{R}(s, a, s') = \begin{cases} +\texttt{MAXSTEPS}, & \text{the goal is reached} \\ -1, & \text{otherwise} \end{cases}$$

**LavaCrossing-Easy/Hard**  The example levels are illustrated in Figure C.6a and Figure C.6b. The task is to navigate toward the goal (green block) while avoiding the lava (orange block).

$$\mathcal{R}(s, a, s') = \begin{cases} +\texttt{MAXSTEPS}, & \text{the goal is reached} \\ -\texttt{MAXSTEPS}, & \text{lava hit} \\ -1, & \text{otherwise} \end{cases}$$

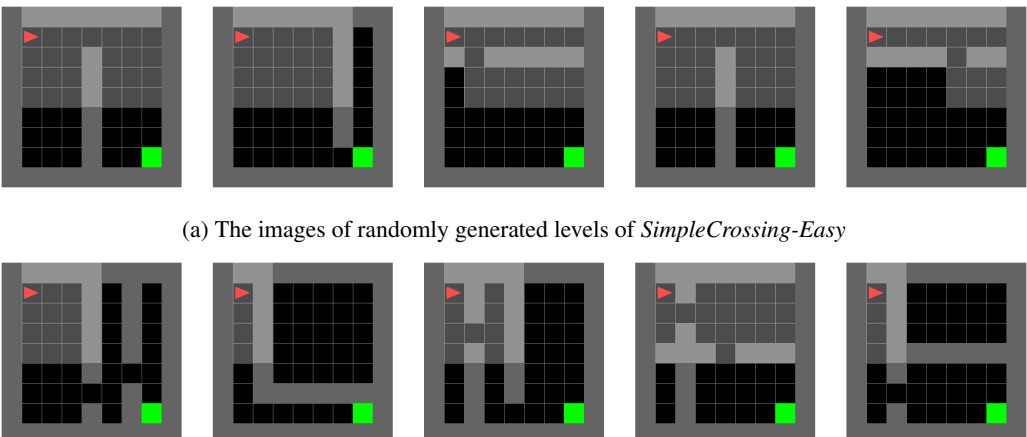

(a) The images of randomly generated levels of *SimpleCrossing-Easy*

(b) The images of randomly generated levels of *SimpleCrossing-Hard*

Figure C.5

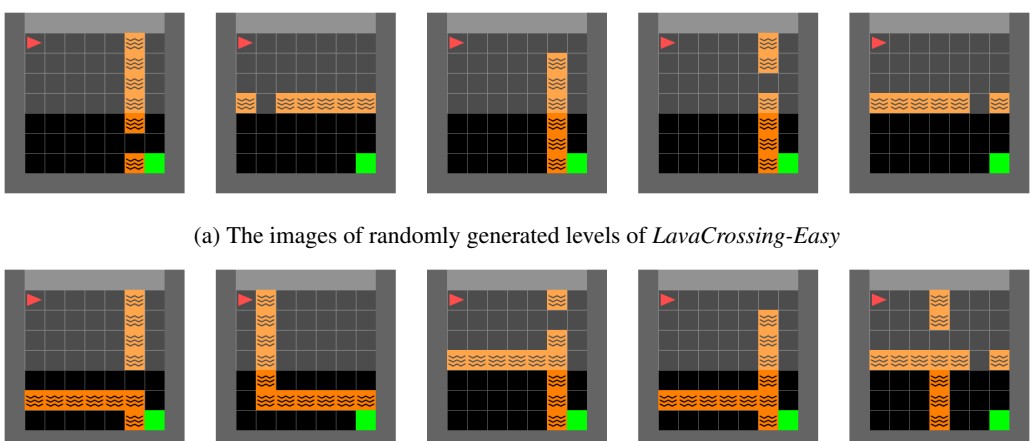

(a) The images of randomly generated levels of *LavaCrossing-Easy*

(b) The images of randomly generated levels of *LavaCrossing-Hard*

Figure C.6

### C.3 SOKOBAN

The implementation can be found in (Schrader, 2018). We resize each RGB image state observation to $84 \times 84$.

## D AVERAGE $Q$-VALUES IN *Sokoban*

We compute the normalized difference between the learned $Q$ values and the maximum return value in the environment: $\frac{\mathbb{E}_{s,a}[Q(s,a)] - R_{\max}}{R_{\max}}$, where $R_{\max}$ is the maximum return the agent can get in the task, and $\mathbb{E}_{s,a}[Q(s,a)]$ is the average $Q$ values during training. Figure D.7 shows the average $Q$-values in *Sokoban*.

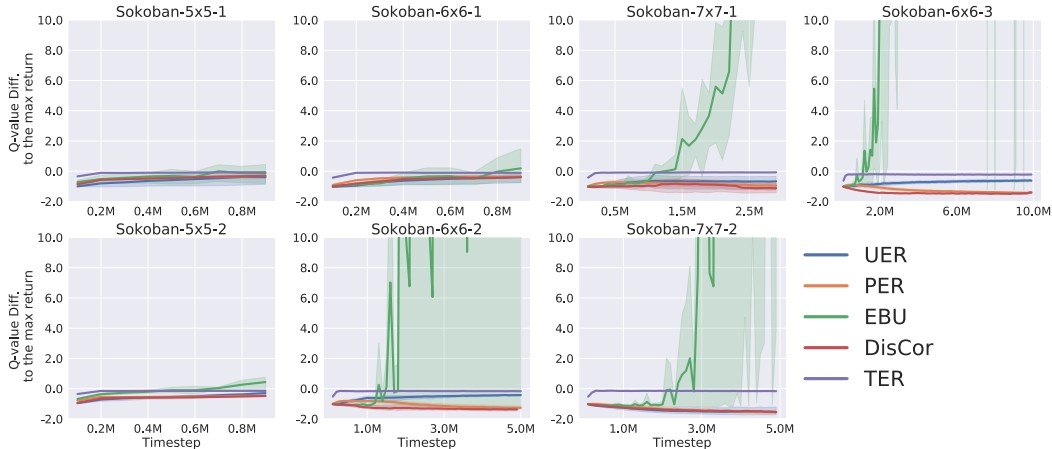

Figure D.7: EBU tends to over-estimate the $Q$-values in *Sokoban*. UER, PER, and DisCor tend to under-estimate.

## E HYPERPARAMETER SEARCH FOR EBU

As the original paper of EBU (Lee et al., 2019) did not have the best-fit $\beta$ for the benchmarks we considered in Section 5.3. Figure E.8 shows that the influence of $\beta$ is not significant. Thus, we follow the best-fit parameter reported in (Lee et al., 2019) and select $\beta = 0.5$ for all of our experiments.

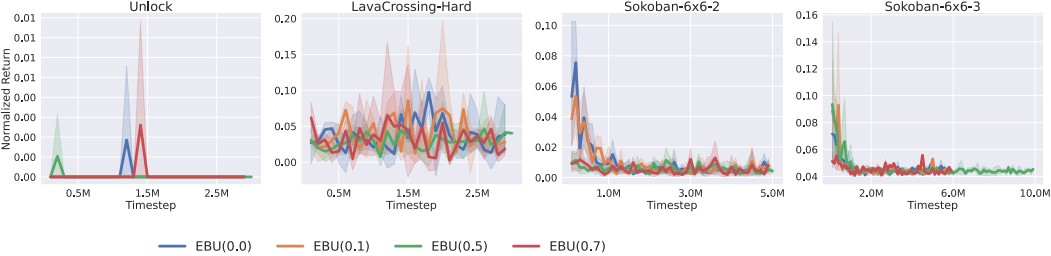

Figure E.8: Performance of EBU with different $\beta$

## F STOCHASTIC ENVIRONMENTS EXPERIMENTS

We examine the performance of TER under stochastic transition dynamics in *LavaCrossing-Hard*. The stochasticity is realized by replacing the agent's action with a random action in 10% of steps in an episode. Figure F.9 shows that this stochasticity does not overshadow the advantage of TER.

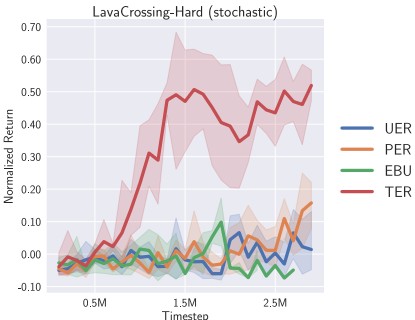

Figure F.9: Performance in stochastic environments

## G   PSEUDO TERMINAL STATE EXPERIMENTS

We further demonstrate an extension of TER to environments without terminal states. We test TER in the modified *SimpleCrossing-Hard* and *LavaCrossing-Hard* where the episode does not terminate when the agent reaches the goal. An episode terminates only when the time is up or entering lava. The agent gets +MAXSTEPS reward when entering the goal, −MAXSTEPS reward when exiting the goal or entering lava, 0 reward when staying at the goal, −1 reward otherwise. For TER, we construct "pseudo terminal states" for reverse sweeping. We keep track of the moving average of the accumulated rewards at a vertex, sample vertices proportionally to their accumulated rewards, and start reverse sweeping from the sampled vertices. The sampled vertices are regarded as "pseudo terminal states". The accumulated rewards of, for example, a vertex $v$ is defined as

$$U(v) := \frac{\sum_{s_t^i \in S(v)} G(s_t^i)}{|S(v)|}$$

where $s_t^i$ denotes the state at timestep $t$ in episode $i$, $G(s_t^i)$ corresponds to the accumulated rewards from the beginning of episode $i$ to timestep $t$, $S(v) := \{(s_t^i \in \Omega \mid v = \phi(s_t^i)\}$ is the set of states mapped to vertex $v$, and $\Omega$ is the replay buffer. The probability of choosing a vertex $v$ is defined as

$$P(v) := \frac{\exp(U(v)/\kappa)}{\sum_{v \in V} \exp(U(v)/\kappa)},$$

where $\kappa$ denotes the temperature (we set $\kappa = 0.01$) of a Boltzmann distribution. Figure G.10 shows that the pseudo terminal state scheme enables TER to outperform the baselines, suggesting that pseudo terminal states approach could be a promising alternative of terminal states. We also find that the baselines' performance drop in the non-terminal version of *SimpleCrossing-Hard* and *LavaCrossing-Hard*. We hypothesize that it is because in the non-terminal version, the agent could accidentally exit the goal after reaching, so the learning is harder than the original environment. Moreover, this fact makes learning more challenging in *LavaCrossing-Hard* since the goal is near lava and the agent could accidentally hit lava and end up with a large negative reward.

## H   DOES BATCH MIXING AID EBU?

To investigate if the performance gap between EBU and TER results from batch mixing, we incorporate batch mixing with EBU and compare it with TER. We set the batch mixing ratio $\eta$ same as that used in TER for every environment. Figure H.11 shows that batch mixing does not improve EBU's performance, which suggests that batch mixing is not a critical reason for TER to outperform EBU.

## I   DOES FILTERING OUT TIMEOUT STATES IMPROVES EBU?

One possibility that TER outperforms EBU is the difference in the starting states of reverse replaying. TER only starts from the terminal states, while EBU initiates backtracking from both terminal

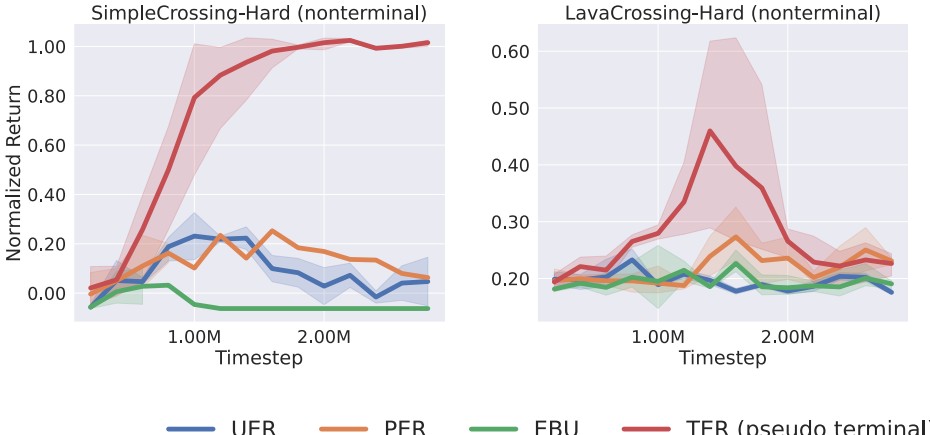

Figure G.10: Performance in environments without terminal states

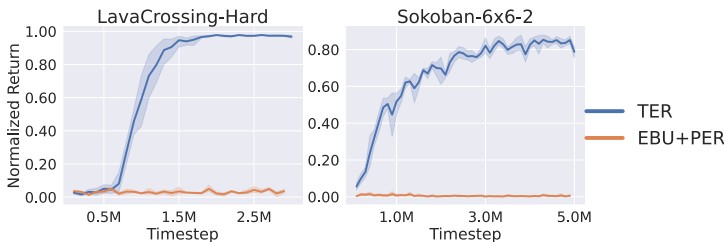

Figure H.11: Incorporating batch mixing with EBU does not aid EBU.

states and timeout states (i.e., the states that trigger resets because of running out of a specified time limit). We then let EBU starts only from the terminal states and compares it again with TER. Figure I.12 shows that *EBU(no_timeout)* did not mitigate the performance gap, which suggesting the choices of starting states of reverse replaying is not crucial.

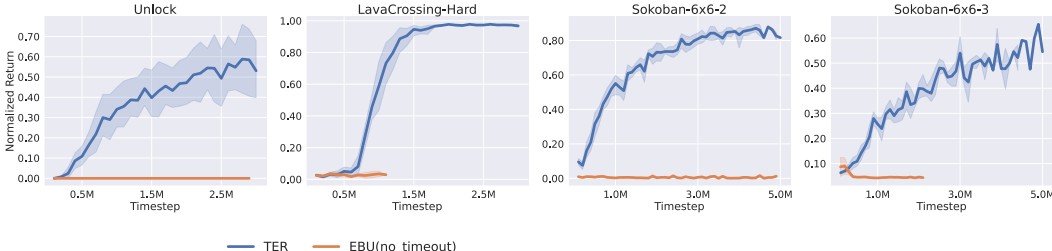

Figure I.12: Filtering out timeout states does not help EBU match TER's performance.

## J    HOW DOES DIMENSION OF $d$ AFFECT TER?

Since we rely on random projection to compress the states and stitch common states in different trajectories, it is interesting to see if the choice of dimension affects the performance. Let $\mathcal{Z} := \mathbb{R}^d$. Figure J.13 surprisingly shows that TER is insensitive to the dimension of the vectors after the random projection. We found that even small variations in images lead to huge difference after random projection. This might explain why even TER($|\mathcal{Z}| = 1$) can distinguish each state. Note that as we do not rely on the distances in the space $\mathcal{Z}$ to determine the neighbor vertices, it is acceptable that a pair of state and successor state $(s, s')$ are distant in the space $\mathcal{Z}$ after random projection.

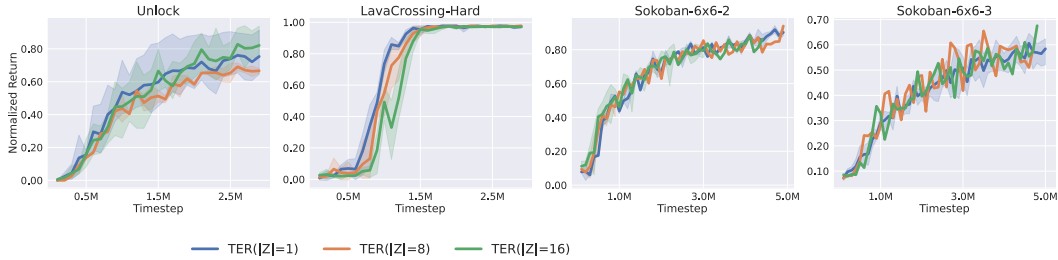

Figure J.13: TER is insensitive to the choice of dimension of $\mathcal{Z}$.

## K COMPARISON WITH STATE-OF-THE-ART MODEL-BASED RL

As van Hasselt et al. (2019) shows that model-free RL using experience replay can be competitive with model-based RL using imaginary data, we are interested whether TER can match or surpass such model-based RL methods. We compare TER with state-of-the-art model-based RL, DreamerV2 (Hafner et al., 2020). We implemented TER with double DQN as we mentioned in Section 5.1. Figure K.14 shows that TER can match DreamerV2's performance at convergence in 3 out of 4 tasks, albeit DreamerV2 has the luxury of access to imaginary data and more sophisticated models. DreamerV2 learns a large environment dynamics model and a policy with the latent representations of the learned dynamics model. The policy is trained by imaginary data generated by the dynamics model. Generating imaginary data entails expensive computations and lots of wallclock time. Also, if the environment dynamics model is discontinuous or contact-rich in nature, it is difficult to fit a good neural network model for generating imaginary data. In such a case, model-free methods can be preferable to model-based methods.

Additionally, the architecture of DreamerV2 is more complex and harder to tune than TER. TER consists of the following components:

$$\textbf{Q-network: } Q(s,a) = Q_\theta(s,a)$$
$$\textbf{Random projection matrix: } \phi(s) = \mathbf{M}s,$$

where $\theta$ denotes the weight of the Q-network, while DreamerV2 comprises the following modules:

$$\textbf{Recurrent model: } h_t = f_\phi(h_{t-1}, z_{t-1}, a_{t-1})$$
$$\textbf{Representation model: } z_t \sim q_\phi(z_t|h_t, x_t)$$
$$\textbf{Transition predictor: } \hat{z}_t \sim p_\phi(\hat{z}_t, h_t)$$
$$\textbf{Image predictor: } \hat{x}_t \sim p_\phi(\hat{x}_t|h_t, z_t)$$
$$\textbf{Reward predictor: } \hat{r}_t \sim p_\phi(\hat{r}_t|h_t, z_t)$$
$$\textbf{Discount predictor: } \hat{\gamma}_t \sim p_\phi(\hat{\gamma}_t|h_t, z_t)$$
$$\textbf{Actor: } \hat{a}_t \sim p_\psi(\hat{a}_t|\hat{z}_t)$$
$$\textbf{Critic: } v_{\mathcal{E}}(\hat{z}_t) \approx \mathbb{E}_{p_\phi, p_\psi} \Big[ \sum_{r \geq t} \hat{\gamma}^{r-t}\hat{r}_\tau \Big].$$

For the explanations of each notation, please refer to Hafner et al. (2019). Each component in dreamer is a neural network and requires tuning the network architecture. As such, we highlight that TER attains similar performance in the selected domains shown in Figure K.14 with less hyperparameters.

## L ADDITIONAL ATARI EXPERIMENTS

We further evaluate TER in four games selected from *Atari* benchmark where these tasks' objectives are not to reach terminal states. Figure L.15 shows that TER performs similarly with the baselines in these four games, which suggests that violating the terminal state reaching assumption in the task's objective does not impede TER's performance though the agent cannot benefit from TER. Exciting future work is to discover promising states other than terminal states to start to reverse sweeping for general RL tasks.

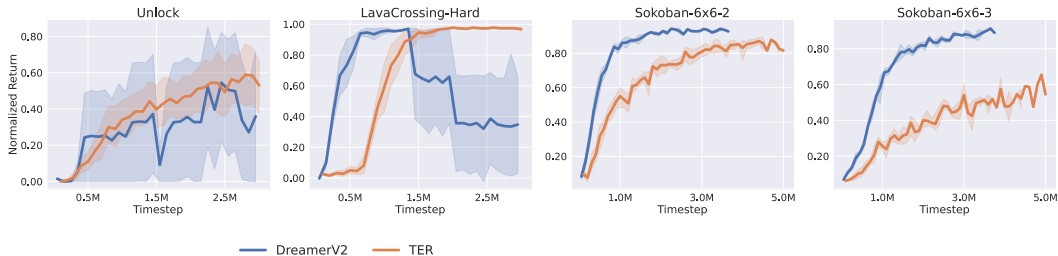

Figure K.14: Comparison between TER and dreamer.

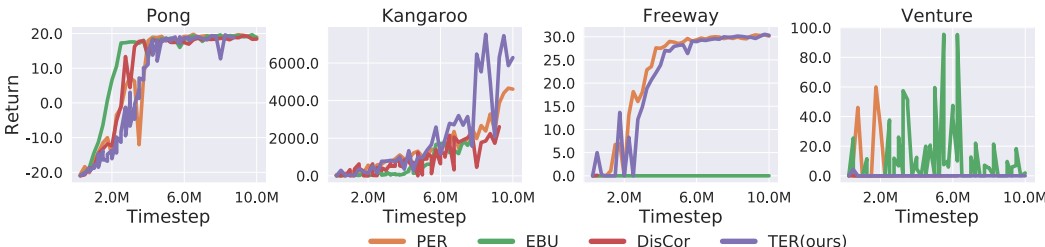

Figure L.15: Learning curves in Atari games.

## M  DISCUSSION OF TER IN CYCLICAL MDPS

We discuss the rationale that TER can work in cyclical MDPs. TER improves the $Q$-learning update efficiency because we update the $Q$-values at dependent successors states before the current state's $Q$-values. However, in cyclical MDPs, some states are inter-dependent (i.e., being predecessor and successor mutually), which renders the correct update order unclear. We argue that under some assumptions, the optimal backup order exists even in cyclical MDPs.

### M.1  OPTIMAL UPDATE ORDER OF $Q$-LEARNING EXISTS IN EPISODIC CYCLICAL MDPS

In cyclical MDPs, there could be an inverse action at the successor state $s'$ that moves the agent back to the current state $s$, which causes inter-dependency. For example, for a state-action pair $(s, a)$ and the corresponding successor state $s'$, there could be an inverse action $a'$ that transitions from $s'$ to $s$.

Fortunately, the $Q$-value $Q(s, a)$ of a state-action pair $(s, a)$ is not dependent on the $Q$-values of all actions at the successor state $s'$. Let us recall the $Q$-value update equation

$$Q(s, a) = \mathcal{R}(s, a, s') + \gamma \max_{a'} Q(s', a').$$

Instead, $Q(s, a)$ depends only on the $\max_{a' \in \mathcal{A}} Q(s', a')$ of the successor state $s'$. Let the optimal successor action $a'^{*} = \arg\max_{a' \in \mathcal{A}} Q(s', a')$. All $Q(s, a')$, $a' \neq a'^{*}$ does not affect the $Q(s, a)$ update. We just need to ensure that $Q(s', a'^{*})$ is updated before $Q(s, a)$. Moreover, for any state $s$, we care only about the optimal action $a^{*} = \arg\max_{a \in \mathcal{A}} Q(s, a)$, if we are only interested in finding the optimal policy.

As the $Q$-value at each state-action pair depends only on the max $Q$-value of the successor state, we are able to define an optimal path from any state. Let a rollout path $\omega$ starting from the state $s$ be $[(s_1, a_1) \ldots (s_{H-1}, a_{H-1}) \mid s_1 = s]$, where $H$ is the path length. The rollout path $\omega$ given by the optimal policy are then $[(s_1, a_1^{*}), \ldots (s_{H-1}, a_{H-1}^{*})]$, where $s_i$ denotes the $i^{th}$ state in the path and $a_i^{*}$ the optimal action at state $s_i$.

Under the assumption that the objective is to reach some terminal states, each $\omega$ is acyclic, and merging all $\omega$ given by the optimal policy forms a directed acyclic graph (DAG). This assumption is valid; otherwise, the optimal policy would be a trivial policy endlessly going back and forth between two states since there must be a terminal state in episodic RL settings. Besides, the merged graph must be a DAG; otherwise, it indicates there is an optimal action $a^{*}$ at a state $s$ is an inverse action.

For instance, suppose we have two states $s_A$ and $s_B$ and their corresponding optimal actions $a_A^*$ and $a_B^*$. Let the transition function be $\mathcal{P} : \mathcal{S} \times \mathcal{A} \mapsto \mathcal{S}$. If $s_B = \mathcal{P}(s_A, a_A^*)$ and $s_A = \mathcal{P}(s_B, a_B^*)$, then the optimal policy will lead the agent to move back and forth between $s_B$ and $s_A$. Given that DAG, the optimal update order is established according to (Bertsekas et al., 2000).

## M.2 OPTIMAL UPDATE ORDER IS HARD TO OBTAIN

Though the optimal backup order for the optimal policy for cyclical MDPs exists, we need the optimal actions at each state for deriving this order. However, the optimal policy is unknown before we finish the training. Fortunately, under an admissible assumption that rewards at non-terminal transitions are non-positive, we can exploit the ending state of the optimal policy.

If rewards are non-positive at transitions other than terminal ones, the MDP can be viewed as an unweighted graph. Under the assumption that the task's objective is to reach some terminal goal states, all the paths given by the optimal policy must end at a terminal state. Performing reverse BFS from the terminal states gives rise to the shortest path (i.e., the optimal policy) from terminal states to all other states, thus recovering the optimal backup order. Note that even though we can infer the optimal paths by the terminal states in these environments, training $Q$-function by the experience associated with these paths is still preferable. We can only infer the optimal paths from the existing experience, but training a $Q$-function based on these paths can generalize to other unseen states.

Nevertheless, there is more than one terminal state in the environment in practice, and we do not know which one is the terminal state that the optimal policy will end at. One approximation is to perform reverse sweep from all the terminal states, as TER does. Though the backup order sequences from these reverse sweeps are not necessarily optimal, we reduce the search space for the optimal backup order to the permutations of the terminal states. Otherwise, the search space for the update order is the permutations of all the possible state-action pairs in the environment.

Note that even though the state-action pairs with non-optimal actions are not needed to be updated (as long as their $Q$-values are initialized small enough), we still perform value backups because their $Q$-values could become higher than that of the optimal actions due to function approximation. Performing value backups on these state-action pairs makes sure that these $Q$-values are not over-estimated.

## N DISCUSSION OF TER IN NON-GRID LIKE TASKS

TER is not restricted to grid-like environments. For instance, consider a *LunarLander* task in OpenAI gym (Brockman et al., 2016). The goal is to control the aircraft to land on the planet. Though it is not a grid-like environment, we still can build a graph to maintain predecessors of each state in *LunarLander* for TER.

## O DISCUSSION

**What if an agent is unlikely to visit the same state twice?** TER is likely to degenerate to EBU in this case as the graph will end up with multiple independent paths. A quick fix is to aggregate vertices (i.e., $\phi(s)$ from random projection) by their Euclidean distances. However, the Euclidean distances in the space spanned by randomly projected vectors might not reflect the state similarity with respect to a task. A viable way is to aggregate states based on their features extracted by the value function. For example, suppose we have transitions $(s_1, a_1, s_2)$ and $(s_3, a_3, s_4)$ and let $\psi(s_i) \ \forall i \in \{1, 2, 3, 4\}$ be the states' features in the value function. When we update $Q(s_2, a)$ for any action $a$, we need to update $Q(s_1, a_1)$ as well as $Q(s_3, a_3)$ if $\psi(s_2) \approx \psi(s_4)$. Unfortunately, the graph has to be rebuilt over again whenever the $Q$-function is updated, which leads to expensive computation during online training. An interesting follow-up is to develop a faster online graph update rule that keeps tracking the value function's feature space $\psi$ changes.

**How does TER tackle stochastic transition function?** For each vertex in the graph, we can maintain the visitation counts from each predecessor during exploration in the environment and calculate the visitation probabilities from each predecessor by the visitation counts. Given that probabilities,

we can sample the predecessors according to their visitation probabilities during reverse BFS. On the other hand, we tested TER in an environment with a stochastic transition function. The results shown in Figure F.9 suggest that stochasticity does not overshadow the advantage of TER.

