# OpenReview forum: "Topological Experience Replay"
_ICLR.cc/2022/Conference — ICLR 2022 Poster_

### Official Review · Reviewer_ANXD · 2021-10-25

**Correctness:** 4
**Technical Novelty And Significance:** 3
**Empirical Novelty And Significance:** 4
**Recommendation:** 8
**Confidence:** 4

**Main Review:**

## Strengths
- The paper's presentation is clear and easy to read
- The paper's main idea is simple to understand, and seems to improve performance robustly
- Stating the assumptions of the method is good.
- Rigorous ablations and analysis are carried out to understand how the method improves on prior work

## Weaknesses
- I wouldn't describe MiniGrid as an image-based environment, and while sokoban may be, the number of different images is still quite low. This makes the state-matching for trajectory stitching possible, which it wouldn't be (naively) in higher dimensional spaces. The hashing could address this, but I think it's worth adding that the hashing being good (or having another way of matching states between trajectories) is effectively an assumption of your method
- Due to the assumptions of terminal states being rewarding, the method is limited, unless a proposed way of improving it to apply to more general reward structures was presented.

## Questions
- What view of minigrid are you using? For example is the veiw ego-centric, or does it include the entire map. And is it the RGB rendering of the map, or the tensor representation?
- It would be good to see results on how often state matches occur in general. Especially for these environments with randomly generated levels, it seems unlikely that states would match unless they're from the same generated level.


**Summary Of The Paper:**

This paper proposes a novel experience replay technique motivated by the topology of the collected data. The technique changes the order in which states are sampled for a minibatch Q-function update, sampling states backwards in a breadth-first way through a trajectory graph constructed from the replay buffer. This means that terminal states, which are more likely (by assumption) to receive reward, and don't require bootstrapping for value estimation, are updated first allowing reward from these states to propagate backwards to earlier states quicker. This method shows improved performance on several MiniGrid and Sokoban tasks, outperforming other baseline experience replay methods (PER, EBU, DisCor). They perform extensive ablations on the difference between TER (their method) and EBU, showing why TER outperforms EBU.

**Summary Of The Review:**

Overall I believe the paper is good enough to be accepted. While a few additional experiments would be beneficial to the understanding of the method's performance, the main blocker towards me giving a higher score is the limitation of the method to environments where the terminal state holds most of the reward. I think the method is still interesting, but showing it can be applied in environments without rewarding terminal states and that it still improve performance in these environments would be beneficial.

EDIT: Having seen the additional experiments and clarifications the authors have added, showing that TER could be used in situations where reward is concentrated in states rather than just in terminal states, I raised my score from 6 to 8.

---

> ### Author Response · Authors · 2021-11-18
> **response**
>
> We are glad to see that the reviewer is interested in our method and thinks our paper is easy to follow! We answer the rest of the questions below.
>
> > *Question 1: I wouldn't describe MiniGrid as an image-based environment, and while sokoban may be, the number of different images is still quite low. This makes the state-matching for trajectory stitching possible, which it wouldn't be (naively) in higher dimensional spaces. **The hashing could address this, but I think it's worth adding that the hashing being good (or having another way of matching states between trajectories) is effectively an assumption of your method***
> >
>
> **Answer:** We have updated our paper based on the reviewer's comment. We appreciate the reviewer's feedback on improving the clarity of our paper.
>
> > *Question 2: Due to the assumptions of terminal states being rewarding, the method is limited, unless a proposed way of improving it to apply to more general reward structures was presented.*
> >
>
> **Answer:**
>
> We would like to clarify that we only need the task's objective is to reach terminal states. We did not make assumptions on the reward structures at terminal states (i.e., terminal states can also give negative rewards or zero rewards).
>
> Regarding applicability to the tasks where the objective is not reaching goals, one can construct "pseudo" terminal states by the states of high returns and then start reverse sweeping from "pseudo" ones. To do so, one can keep track of the moving average of accumulated rewards (i.e., episodic returns) ending at each state. This would be a reasonable approximation to the terminal states since the states that accumulate high rewards will propagate values to their predecessors in value backups.
>
> Even though some RL benchmarks might not be goal-reaching tasks, we would also like to emphasize that goal-reaching constitutes an important family of tasks. For example, many robotic tasks, such as pick and place, object re-orientation, navigation are goal-reaching. Unfortunately, goal-reaching tasks are often difficult to solve by typical RL methods due to lack of intermediate reward signals guiding the agent to improve the policy/value function in a correct way.
>
> The importance and the difficulty of goal-reaching tasks motivate us to focus on solving family of tasks. In this paper, we present TER as a proof-of-concept that exploits the states' dependency to expedite the Q-function training. An exciting future avenue is to extend TER to robotic tasks in the real world.
>
> [1] Andrychowicz, Marcin, et al. "Hindsight experience replay.
>
> > *Question 3: What view of minigrid are you using? For example is the veiw ego-centric, or does it include the entire map. And is it the RGB rendering of the map, or the tensor representation?*
> >
>
> **Answer:** We use the RGB rendering of the entire map.
>
> > *Question 4: It would be good to see results on how often state matches occur in general. Especially for these environments with randomly generated levels, it seems unlikely that states would match unless they're from the same generated level.*
> >
>
> **Answer:** Yes, the states from different levels are unlikely to be connected. Our goal was never to connect states across levels, but simply states within each level.

---

### Official Review · Reviewer_t7F3 · 2021-10-28

**Correctness:** 3
**Technical Novelty And Significance:** 2
**Empirical Novelty And Significance:** 3
**Recommendation:** 6
**Confidence:** 5

**Main Review:**

However, I have the following concerns:

Q1. I believe the current version of TER is applied only to goal-reaching tasks where the learning objective is to reach goal states. The authors should clarify it in the abstract and introduction sections, and discuss how to extend TER to general RL tasks.

Q2. I suspect that the hashing method may lead to an intractably large graph $\mathcal{G}$ especially for environments with large state spaces. I think it may be necessary to partition the state space into multiple state clusters to obtain a tractable graph.

Q3. I suspect that inaccurate graphs may lower the Q-learning convergence speed. Does it take a large number of samples to construct an accurate state graph? If so, the TER algorithm requires some warming-up time-steps to collect samples to construct a comparatively accurate state graph, which should be discussed in the main text.

Q4. The authors only consider the construction of undirected graphs, which makes their method not applicable to MDPs where states form directed graphs.


**Summary Of The Paper:**

The authors propose a topological experience replay (TER) method to perform Q value updates in a reverse sweep style. A variety of empirical results demonstrate that TER improves the Q-learning convergence speed significantly. The paper is well-written and well-organized, and the motivation is clear.

**Summary Of The Review:**

Although the paper is well written, the current version does not convince me to recommend acceptance.

---

> ### Author Response · Authors · 2021-11-18
> **response**
>
> Thanks for the valuable feedback. We address the reviewer's concerns below.
>
> > *Q1. I believe the current version of TER is applied only to goal-reaching tasks where the learning objective is to reach goal states. The authors should clarify it in the abstract and introduction sections, and discuss how to extend TER to general RL tasks.*
> >
>
> **Answer:**
>
> We would like to mention that TER can also be applied to non goal-reaching tasks, but in these scenarios TER will simply match the performance of prior methods. We would also like to emphasize that goal-reaching constitutes an important family of tasks. For example, many robotic tasks, such as pick and place, object re-orientation, navigation are goal-reaching. Unfortunately, goal-reaching tasks are often difficult to solve by typical RL methods due to lack of intermediate reward signals guiding the agent to improve the policy/value function in a correct way.
>
> The importance and the difficulty of goal-reaching tasks motivate us to focus on solving this family of tasks. Current experiments show that TER exploits the states' dependency to expedite the Q-function training. An exciting future avenue is to extend TER to robotic tasks in the real world.
>
> We thank the reviewer for the suggestion and will update the writing to reflect when TER improves performance.
>
> Regarding applicability to the tasks where the objective is not reaching goals, one can construct "pseudo" terminal states by the states of high returns and then start reverse sweeping from "pseudo" ones. To do so, one can keep track of the moving average of accumulated rewards (i.e., episodic returns) ending at each state. This would be a reasonable approximation to the terminal states since the states that accumulate high rewards will propagate values to their predecessors in value backups.
>
> [1] Andrychowicz, Marcin, et al. "Hindsight experience replay."
>
> > *Q2. I suspect that the hashing method may lead to an intractably large graph G especially for environments with large state spaces. I think it may be necessary to partition the state space into multiple state clusters to obtain a tractable graph.*
> >
>
> **Answer:** This is a valid concern. We overcome this issue by pruning the graph nodes when the graph size exceeds a given threshold, similar to how old data is deleted in when replay buffer capacity is reached in other methods. Thus, the graph size does not  grow endlessly. The details of the pruning implementation are provided in Section A.5 of the supplementary material.
>
> > *Q3. I suspect that inaccurate graphs may lower the Q-learning convergence speed. Does it take a large number of samples to construct an accurate state graph? If so, the TER algorithm requires some warming-up time-steps to collect samples to construct a comparatively accurate state graph, which should be discussed in the main text.*
> >
>
> **Answer:** TER does not require a large number of samples to build a graph since we use random projection to map a state observation to a node. Random projection does not require training.  As a result, TER does not need an additional warm-up period other than the standard warm-up period required by DQN.
>
> > *Q4. The authors only consider the construction of undirected graphs, which makes their method not applicable to MDPs where states form directed graphs.*
> >
>
> **Answer:** The graph we build is directed. For example, given a transition $(s, a, s', r)$, we will add a directed edge $(\phi(s) \rightarrow \phi(s'))$ to the graph.

---

> > ### Comment · Reviewer_t7F3 · 2021-11-23
> > **Response**
> >
> > Thanks for the response that addressed most of my concerns. However, I still think applicability to general tasks is important. I suggest the authors add the approach to extending TER to general tasks and more experiments to evaluate the TER performance over some general tasks.

---

> > > ### Author Response · Authors · 2021-11-25
> > > **response**
> > >
> > > We thank the reviewer for the reply. We have added additional experiments to demonstrate the possibility of extending TER to non goal-reaching tasks in Section G.
> > >
> > > We tested TER in the modified $\textit{SimpleCrossing-Hard}$ and $\textit{LavaCrossing-Hard}$, where the task's objective is to "stay" at the goal state. Note that these tasks differ from goal-reaching tasks because there are no terminal states indicating success of goal completion.
> > >
> > > In the modified environments, episodes do not terminate when the agent reaches the goal. An episode terminates only when the time is up or the agent enters lava. The agent gets $\texttt{+MAXSTEPS}$  reward when entering the goal, $\texttt{-MAXSTEPS}$ reward when exiting the goal or entering lava, $0$ reward when staying at the goal, $-1$ reward otherwise. For TER, we construct "pseudo-terminal states" for reverse sweeping. We keep track of the moving average of the accumulated rewards at a vertex, sample vertices proportionally to their accumulated rewards, and start reverse sweeping from the sampled vertices. The sampled vertices are regarded as "pseudo-terminal states".
> > >
> > > Figure G.10 shows that the pseudo-terminal state scheme enables TER to outperform the baselines, suggesting that the pseudo-terminal state's approac could be a feasible alternative to terminal states. We also find that the baselines' performance drops in the non-terminal version of $\textit{SimpleCrossing-Hard}$ and $\textit{LavaCrossing-Hard}$. We hypothesize that it is because, in non-terminal environments, the agent could accidentally exit the goal after reaching it, so the learning is harder than in the original environment. Moreover, this fact makes learning more challenging in $\textit{LavaCrossing-Hard}$ since the goal is near lava and the agent could accidentally hit lava and end up with a large negative reward.
> > >
> > > We hope these experiments are helpful on demonstrating the feasibility of our method's application to more general RL tasks.

---

> > > > ### Comment · Reviewer_t7F3 · 2021-11-25
> > > > **The main concern is addressed**
> > > >
> > > > Thanks for the additional experiments and discussion that address my main concern. Please add them to the main text of the final version. I would like to raise the score.
> > > >
> > > > I agree with Reviewer ANXD. The experiments show promise in extending TER to general tasks. However, I think the pseudo-terminal states may have significant effect on the TER performance especially for complex tasks. I suggest, in the final version, the authors discuss difficulties for constructing pseudo-terminal states and the possible approach to addressing them.

---

### Official Review · Reviewer_mMYf · 2021-10-30

**Correctness:** 3
**Technical Novelty And Significance:** 3
**Empirical Novelty And Significance:** 2
**Recommendation:** 8
**Confidence:** 4

**Main Review:**

Strengths
-------------
-Finding more sampling efficient ways to sample from a replay buffer is an important avenue of research.

-The authors are clear and open about the details of the implementation.

-There are extensive experiments that shine light on why the method works compared to previous approaches. It also showcases limitations.

Weaknesses
-------------
-The experiments are somewhat limited to goal oriented tasks.

-The number of random seeds is a bit low.


Details
----------
Given the fact that replay buffers can be used with off-policy learning to accelerate model free reinforcement learning methods, it is surprising there is so little research around this topic. Reverse sweeping is intuitive and compelling, yet it is not straightforward to extend it to high dimensional data. Although the authors more or less limit the scope of applicability of their method to grid-like environments, it stills shows that such inductive bias can be useful for complex environments. This is somewhat acknowledged in the "Assumptions" section, however the assumption on being able to revisit the same state twice is left to the conclusion, which is questionable. I think it would be better for the setting to be clear from the beginning, as we can better evaluate the generality, limitations and future work throughout reading the paper.

The details about graph construction are definitely appreciated. The "Overview" itself seems a bit lengthy and prehaps confusing by going sometimes into too much details. Regarding BFS, how many predecessors are sampled? If I understand correctly, there are multiple transition for each edge, but only one state for each node? Or is it possible that \phi maps two states to the same low dimensional space? Regarding hashing and random projections, it is almost un-intuitive for me that this would work. Could the authors elaborate on why this works in this setting, as well as elaborate on what setting would we see this choice be a bottleneck? Did the authors consider using neural networks to project the state, perhaps in a pre-trained manner?

Regarding the difference between EBU and TER, the authors mention that it is likely due to tracjectory stitching. Although I can probably guess what this means, it is never clearly mentioned in the paper what this refers to precisely. In TER(single pred, $\eta=0$), when a single predecessor is designated, does this still leave open the possibility to having multiple trajectories in the edge in-between?
The experiments in Figure 3 are pretty good. I would suggest clearly mentioning the values of $\eta$. I would also suggest augmenting the number of random seeds to 10. In the same spirit of reproducibility, I think it would be great to present the kind of hyper parameter search that was done for TER.

Regarding related work, the authors do cite (Lin, 1992), however they do not mention that within this work the author suggests and uses backward replay (i.e. a sequence is replayed in a backward order for updates), which highlights that the idea of reverse sweep is common itself. As the paper itself considers building a graph, references to recent work are somewhat limited. For example Jiang et al. 2021 and  Klissarov et al. 2020 very different method for creating the graph. The latter work is perhaps more related to this work in the sense that it seeks better sample efficiency within a single task. In general, the construction of the graph is a bottleneck for RL methods, and citing relevant literature in this space is important as it can accelerate future research.



**Summary Of The Paper:**

The paper proposes a new sampling strategy to improve the sample efficiency of Q-learning based methods. Bootstrapping being a fundamental characteristic of TD methods, the accuracy of the next state's value is critical in updating the current state. Previous work preform a similar observation, leading to methods such as backward value iteration. However, performing reverse sweeps is challenging in high dimensional tasks. As such, the authors propose to iteratively construct a graph representing the underlying MDP on which reverse sampling is executed. The authors report considerable improvement on grid-like tasks when compared to baselines that propose sampling schemes. The authors also provide numerous additional experiments attempting to elucidate the merits and limitations of the approach.

**Summary Of The Review:**

The authors provide a clear explanation of their method, code and implementation details. Although the experimental setup is a bit limited,  the results look good and the proposed approach has the potential to improve future work in this space. Moreover, many additional experiments are proposed to shine light upon the proposed method.

---

> ### Author Response · Authors · 2021-11-18
> **response (1/2)**
>
> We thank the reviewer for the valuable feedback. We answer the unaddressed questions below.
>
> > *I think it would be better for the setting to be clear from the beginning, as we can better evaluate the generality, limitations and future work throughout reading the paper.*
> >
>
> **Answer:** We thank the reviewer's suggestion! We have modified our presentation structure in the updated manuscript (marked in magenta).
>
> > *The "Overview" itself seems a bit lengthy and prehaps confusing by going sometimes into too much details.*
> >
>
> **Answer:** Thanks for the suggestions on improving our writing! We have condensed the text in the overview section in the updated manuscript.
>
> > *Regarding BFS, how many predecessors are sampled?*
> >
>
> **Answer:** We sample at most 4 predecessors at each node.
>
> > *If I understand correctly, there are multiple transition for each edge, but only one state for each node? Or is it possible that \phi maps two states to the same low dimensional space?*
> >
>
> **Answer:**  There are multiple states at each node. Two states are mapped to the same node as long as their encoded vectors $\phi(s)$ are the same.
>
> > *Regarding hashing and random projections, it is almost un-intuitive for me that this would work. Could the authors elaborate on why this works in this setting, as well as elaborate on what setting would we see this choice be a bottleneck?*
> >
>
> **Answer**:  First, we like to clarify that our **random projection is not a hash function**, but an **encoding function that transforms high-dimensional states to dimensional vectors**. The low-dimensional vectors serve as the keys of the hash-table, and the table uses hash-function built-in python. We are sorry for the confusion and have updated the manuscript!
>
> It is indeed surprising that random projection can produce the keys that distinguish different states. Prior works [1,2,3] also have shown (quite surprisingly) that random projection can make a reasonable graph or episodic memory in Atari.
>
> [1] Zhu, Guangxiang, et al. "Episodic reinforcement learning with associative memory." (2020).
>
> [2] Lin, Zichuan, et al. "Episodic memory deep Q-networks." *arXiv preprint arXiv:1805.07603* (2018).
>
> [3] Blundell, Charles, et al. "Model-free episodic control." *arXiv preprint arXiv:1606.04460* (2016).
>
> We hypothesize this is because the action spaces are discrete in Minigrid, Sokoban, and Atari, so the number of distinct images is finite. When the action space is continuous, encoding nodes by random projection might not be able to find the same state twice since the number of distinct images will be nearly infinite. In this case, TER will match EBU's performance since the graph degenerates to multiple independent paths (each path is one episode).
>
> To go beyond EBU in continuous state space, in the future works, we can train a neural network that embeds similar states to the same node in the graph, so one can build a graph in a continuous state space. For example, one can use VQ-VAE [4] to learn a discrete representation to represent a node. Another possible direction is to use [5] to learn a state embedding that maps temporally close states together. Though making a more complicated state-to-vertex embedding function can be helpful in more complex domains, we use random projection in our paper since learning a better embedding function is beyond the focus of our paper.
>
> [4] Oord, Aaron van den, Oriol Vinyals, and Koray Kavukcuoglu. "Neural discrete representation learning." *arXiv preprint arXiv:1711.00937* (2017).
>
> [5] Savinov, Nikolay, Alexey Dosovitskiy, and Vladlen Koltun. "Semi-parametric topological memory for navigation." *arXiv preprint arXiv:1803.00653* (2018).
>
> > *Did the authors consider using neural networks to project the state, perhaps in a pre-trained manner?*
> >
>
> **Answer:** This is a great suggestion. A pre-trained neural network could encode states in a more meaningful way, which can solve the problem of graph building in the continuous state space. We will definitely incorporate this suggestion into our future work.
>
> > *Regarding the difference between EBU and TER, the authors mention that it is likely due to tracjectory stitching. Although I can probably guess what this means, it is never clearly mentioned in the paper what this refers to precisely.*
> >
>
> **Answer:** We have improved the explanation and wording in Section 5.6. "stitching" means finding a "common" state between two trajectories and connecting these two trajectories as a graph using this common state.

---

> ### Author Response · Authors · 2021-11-18
> **response (2/2)**
>
>
> > *In TER(single pred, η=0), when a single predecessor is designated, does this still leave open the possibility to having multiple trajectories in the edge in-between?*
> >
>
> **Answer:** "single pred" indicates that we sample only one predecessor during BFS, but the graph still stores multiple predecessors at a node.
>
> > *The experiments in Figure 3 are pretty good. I would suggest clearly mentioning the values of η.*
> >
>
> **Answer:** We have moved the $\eta$ settings from the supplementary material to the main text in the updated manuscript.
>
> > *I would also suggest augmenting the number of random seeds to 10. In the same spirit of reproducibility,*
> >
>
> **Answer:** We will be running more random seeds in the next version of manuscript during rebuttal.
>
> > *I think it would be great to present the kind of hyper parameter search that was done for TER.*
> >
>
> **Answer:**  In the current version, we presented the search results on TER's hyperparameters (e.g., mixing ratio $\eta$ (Figure 5), random projection dimension (Figure I.12)).
>
> > *however they do not mention that within this work the author suggests and uses backward replay (i.e. a sequence is replayed in a backward order for updates), which highlights that the idea of reverse sweep is common itself.*
> >
>
> **Answer:** We appreciate that the reviewer points out this. We have added this description to the related works section.
>
> > *Jiang et al. 2021* ([https://arxiv.org/pdf/2102.04220.pdf](https://arxiv.org/pdf/2102.04220.pdf))
> >
>
> **Answer:** They encode the spatial relational structures in the states by graph neural networks and replace the CNN with the graph neural network in DQN.
>
> > *Klissarov et al. 2020* [https://deepmind.com/research/publications/2020/Reward-Propagation-Using-Graph-Convolutional-Networks](https://deepmind.com/research/publications/2020/Reward-Propagation-Using-Graph-Convolutional-Networks)
> >
>
> **Answer:** They learned a graph neural network to generate intrinsic rewards for RL agents to aid exploration in sparse reward tasks.
>
> DIfferent from the *Jiang et al. 2021* and *Klissarov et al. 2020*, our method builds a graph to guide the replay process, instead of modifying the network architecture and reshaping reward functions.

---

### Official Review · Reviewer_JmAq · 2021-10-31

**Correctness:** 3
**Technical Novelty And Significance:** 2
**Empirical Novelty And Significance:** 2
**Recommendation:** 5
**Confidence:** 4

**Main Review:**

This paper clearly tackles an important problem to a currently relevant class of algorithms in the community. However, the proposed solution is completely tailored to a particular type of reinforcement learning problem: episodic, goal reaching problems, where the observed reward signal is zero in every transition but the final one. This drastically reduces the applicability of the proposed method. Some important components/assumptions of the overall approach are also not properly discussed/evaluated. Moreover, this is a paper that touches on one of the most fundamental aspects of deep RL solutions, but it justifies itself only on a small set of domains that are not necessarily the domains in which deep RL algorithms (and their replay buffers) were originally proposed and evaluated. This raises more questions about the real feasibility of the propose approach. The text could also be improved for precision.

First and foremost, it seems to me that TER is only effective in RL problems in which the agent observes a reward signal of value zero until success. This seems fairly limiting to me. How can one use TER when the RL agent is tackling problems in which the reward function doesn’t have this structure? Robotics, for example, often uses a dense reward function which is a function of the distance between the robot and the goal. This is reflected in reward functions such as those used in simulators such as MuJoCo, for example. Other simulators have different patterns, but even Atari games that became notorious for their sparsity of rewards (e.g., Montezuma’s Revenge) are not problems in which only one non-zero reward signal is observed. How much TER is a general approach and how much is it a method to leverage very specific features of a subclass of problems?

Some of the assumptions apparently being made also seem quite limiting. Specifically, experience replay buffers are often used for deep RL, when the state space is so large that function approximation is necessary. In that case, it is not clear how often one observes common states in different trajectories. Again, this is a fairly limiting assumption, not to mention that things such as distractors and randomness may even make two equivalent states look differently. The discussion at the end tries to dismiss this suggesting “quick fixes”, but these are never evaluated. The same applies to the distinction between stochastic/deterministic environment. I don’t think the single experiment n Figure F.9 is enough to justify this is not an issue.

Related to this, I also have questions about the proposed method itself. For example, how important is the hashing component of TER? For distinct states it would obviously not lead to quick retrieval, but leaving this aside, how are collisions dealt with? Doesn’t the hash function itself matters? Also, the main justification for using an experience replay buffer in DQN was to decorrelate samples. Now, samples are not decor related, but the backups are heavily tied to the trajectory itself. I imagine this can have a horrible interaction with modern neural network training schemes, generating a lot of instability in non-toy domains.

The text has several issues in terms of precision or correctness. For example:

- “Off-policy algorithms are more data efficient than their on-policy counterparts by learning from the experience replay (Lin, 1992).” This doesn’t seem to be an obvious fact. What off-policy methods? What on-policy methods? What is the role things such as eligibility traces and importance sampling can have in this analysis? What about n-step methods? They are quite effective for whole trajectories.
- “The key ingredient in off-policy methods is Q-learning (Watkins & Dayan, 1992) that learns a Q-function to predict the expected future sum of rewards (i.e., return) at a state.” I’m not sure what this sentence means. Is Q-learning a key ingredient? Isn’t it a method? Learning Q-functions is ominous to most RL algorithms, off-policy and on-policy.
- “Let the agent receive a reward when it reaches the goal state (labeled as G), but not at any other state.” By definition, the agent receives a reward at every time step in an MDP.
- “We ran each experiment with 5 different random seeds and reported the mean and 95%-confidence interval on the learning curves.” How was the confidence interval computed with only 5 samples?


**Summary Of The Paper:**

This paper proposes a new sampling schedule for the experience replay buffer often used by reinforcement learning algorithms. The proposed approach is to perform value backups via bread-first search, starting from the set of terminal states and then moving backwards. The paper proposes a way to building such graph from sampled transitions and it shows, empirically, that the proposed approach (TER) performs better than other methods in a specific class of problems.

**Summary Of The Review:**

This paper clearly tackles an important problem to a currently relevant class of algorithms in the community. However, the proposed solution is completely tailored to a particular type of reinforcement learning problem: episodic, goal reaching problems, where the observed reward signal is zero in every transition but the final one. This drastically reduces the applicability of the proposed method. Some important components/assumptions of the overall approach are also not properly discussed/evaluated. Moreover, this is a paper that touches on one of the most fundamental aspects of deep RL solutions, but it justifies itself only on a small set of domains that are not necessarily the domains in which deep RL algorithms (and their replay buffers) were originally proposed and evaluated. This raises more questions about the real feasibility of the propose approach. The text could also be improved for precision.

---

> ### Author Response · Authors · 2021-11-18
> **response (1/3)**
>
> We appreciate the reviewer's valuable feedback. Our responses to the rest of the questions are as follows.
> > *Question 1: First and foremost, it seems to me that TER is only effective in RL problems in which the agent observes a reward signal of value zero until success. This seems fairly limiting to me. How can one use TER when the RL agent is tackling problems in which the reward function doesn’t have this structure?*
> >
>
> **Answer**:
>
> We apologize for the unclarity: our method is **not restricted to binary reward functions** (i.e., zero or one reward). E.g.. in our Sokoban and Minigrid experiments, the rewards are not binary. In Sokoban, the agent gets rewards +0.1 when the agent pushes one box into the correct position. In Minigrid, a large negative reward is given when the agent enters lava. The only required assumption is that the task's objective is to reach terminal states. The reward specification is detailed in Section C in the supplementary material.
>
>
>
> > *Question 2: How much TER is a general approach and how much is it a method to leverage very specific features of a subclass of problems?*
> >
>
> **Answer**:
>
> TER can be applied to general RL tasks and improves performance for goal reaching tasks. We have shown that TER improves the sample efficiency of DQN in goal-reaching tasks, and matches the prior experience replay methods in non goal-reaching tasks (see Section K in the supplementary material for the experimental results in non goal-reaching games).
>
> > Question 3: **Some of the assumptions apparently being made also seem quite limiting.** Specifically, experience replay buffers are often used for deep RL, when the state space is so large that function approximation is necessary. In that case, **it is not clear how often one observes common states in different trajectories**. Again, this is a fairly limiting assumption, not to mention that things such as **distractors and randomness** may even make two equivalent states look differently. The discussion at the end tries to dismiss this suggesting “quick fixes”, but these are never evaluated. The same applies to the distinction between **stochastic/deterministic environmen**t. I don’t think the single experiment n Figure F.9 is enough to justify this is not an issue.*
> >
>
> **Answer**:
>
> **Visiting the same state twice:** TER indeed benefits from the assumption of visiting the same state twice, but when this assumption does not hold true, our method will still outperform uniform and prioritized experience replay, and is equivalent to EBU. It is equivalent to EBU because if there is no common state between two trajectories, the graph ends up with multiple independent paths instead of a densely connected graph. But even in these paths, the order of Q-learning updates matters, which is why our method will outperform uniform / prioritized experience replay.
>
> **Distractor/Randomness:** We acknowledge the reviewer's question. Distractor/randomness would affect the graph building in a way that two equivalent states could be mapped to differen nodes under distractor. However, distractor and randomness can also damage the original DQN algorithm and are indeed critical problems in deep RL. We would like to kindly bring the reviewer's attention to that this distractor/randomness issue is not the focus of our paper. We aim to present a new perspective and a proof-of-concept of using states' dependency to improve the Q-learning's efficiency in goal-reaching tasks. The core idea of updating the Q-function by reverse sweep ordering is valid regardless of how we build the graph.  For future works on environments with distractor, one could use, for example, [1], to learn the task-informed state features, so one can map two equiavalent states to the same node under distraction.
>
> [1] Fu, Xiang, et al. "Learning task informed abstractions." *International Conference on Machine Learning*. PMLR, 2021.
>
> ***Stochastic/Deterministic environment:*** We understand the reviewer's concern and appreciate the reviewer for reading our paper thoroughly. Nevertheless, we would kindly bring the reviewer's attention to the significance of the experiment in Figure F.9. In Figure F.9, LavaCrossing is the environment where the agent is most likely to be affected by stochasticity. If the agent is overly optimistic about the expected reward on a transition near lava, it is likely that the agent will overlook the risks of getting close to lava and end up with pretty low returns. As a result, we select this domain to test the influence of stochasticity.

---

> ### Author Response · Authors · 2021-11-18
> **response (2/3)**
>
> > *Question 4: Related to this, I also have questions about the proposed method itself. For example, **how important is the hashing component of TER?***  For distinct states it would obviously not lead to quick retrieval, but leaving this aside, how are collisions dealt with? Doesn’t the hash function itself matters?
> >
>
> **Answer**:
>
> First, we like to clarify that our random projection is not a hash function, but an encoding function that transforms high-dimensional states to low-dimensional vectors. The low-dimensional vectors serve as the keys of the hash-table, and the table uses hash-function built-in python. We are sorry for the confusion and have updated the manuscript.
>
> Hashing is important to TER since hashing enables us to map a state to a vertex in a graph in $O(1)$ time complexity. Fast computation is critical for TER since TER will be mapping from a state to a vertex frequently during training.
>
> Regarding the collision and distinct states, collision is handled by python's dictionary data structure. The distinct states are likely to lead to faster retrieval since their keys would be largely different from the others in the table, so the collision chance could be lower.
>
>
> > Question 5: *Also, the main justification for using an experience **replay buffer in DQN was to decorrelate samples**. **Now, samples are not decor related,** but the backups are heavily tied to the trajectory itself. I imagine this can have a horrible interaction with modern neural network training schemes, generating a lot of instability in non-toy domains.*
> >
>
> **Answer**:
>
> This is a great question. If we were using samples from just a single trajectory, correlation between data points would create instability. However, we use data from multiple trajectories (specifically different BFS search trees originating at different terminal nodes) sampled from the replay buffer in one batch. Even though some transitions are tied to the same trajectory, the batch of training data consists of transitions from multiple different trajectories. Therefore, the sample correlation is reduced.
>
> > *Question 6: “Off-policy algorithms are more data efficient than their on-policy counterparts by learning from the experience replay (Lin, 1992).” This doesn’t seem to be an obvious fact. What off-policy methods? What on-policy methods? What is the role things such as eligibility traces and importance sampling can have in this analysis? What about n-step methods? They are quite effective for whole trajectories.*
> >
>
> **Answer**:
>
> We apologize for the imprecise sentences. We intended to communicate that off-policy methods are likely to have a chance to consume less data than on-policy methods because of the experience replay. We did not attempt to argue that all of the off-policy methods are more efficient than on-policy methods. We have rewritten this sentence in the updated manuscript.
>
> > *Question 7: “The key ingredient in off-policy methods is Q-learning (Watkins & Dayan, 1992) that learns a Q-function to predict the expected future sum of rewards (i.e., return) at a state.” I’m not sure what this sentence means. Is Q-learning a key ingredient? Isn’t it a method? Learning Q-functions is ominous to most RL algorithms, off-policy and on-policy.*
> >
>
> **Answer**:
>
> Sorry for being unclear. We intended to say that learning a Q-function is critical in off-policy methods. We have modified the text in the updated manuscript.
>
> We acknowledge that Q-functions can also be important to some on-policy methods (e.g., SARSA), but because most of the off-policy methods require estimating the Q-values, we regard learning Q-functions as a critical ingredient in off-policy methods. On the other hand, learning Q-functions is **not** a **critical** ingredient in many on-policy methods such as PPO. For example, in PPO,  a state value function (instead of a Q function) is learned to serve as a baseline to reduce the variance of policy gradient.
>
> > *Question 8: “Let the agent receive a reward when it reaches the goal state (labeled as G), but not at any other state.” By definition, the agent receives a reward at every time step in an MDP.*
> >
>
> **Answer**:
>
> We apologize for being imprecise. The agent indeed receives rewards at every step in our toy example shown in Figure 1.  We refer "rewards" to the "positive signal" in our toy example. We intended to say that the "positive signal" (i.e., rewards = +1) is only emitted when the agent reaches the goal state, otherwise, the agent does not receive any **positive** signals (i.e., rewards = -1 or 0). Note that our method **does not** rely on this reward structure. This reward structure is just for illustrating the idea.

---

> ### Author Response · Authors · 2021-11-18
> **response (3/3)**
>
> > *Question 9: “We ran each experiment with 5 different random seeds and reported the mean and 95%-confidence interval on the learning curves.” How was the confidence interval computed with only 5 samples?*
> >
>
> **Answer**:
>
> We use the formula described in [https://en.wikipedia.org/wiki/Confidence_interval](https://en.wikipedia.org/wiki/Confidence_interval). The formula is $CI = \mu \pm  1.96 \dfrac{\sigma}{\sqrt{n}}$, where $\mu$ and $\sigma$ are the mean and the standard deviation of the returns of all the random seeds.

---

> > ### Comment · Reviewer_JmAq · 2021-11-20
> > **Confidence interval computation is wrong**
> >
> > The formula referenced above that uses 1.96 is based on the assumption that the distribution of the data is normal. You cannot guarantee that with 5 samples. I would expect a bootstrapping method to be used, otherwise the computation is likely wrong and underestimates the confidence interval.

---

> > > ### Author Response · Authors · 2021-11-23
> > > **correction (updated the manuscript)**
> > >
> > > Thanks for the reviewer's notice and we have checked the settings of our plotting library again.
> > >
> > > We would like to correct our previous response to confidence interval computation. In fact, we use bootstrapping to compute the confidence interval. We plot the confidence interval using `lineplot` of [seaborn](https://seaborn.pydata.org/) visualization library. We did not notice that the `seaborn` uses bootstrapping as the default confidence interval computation [(reference)](https://seaborn.pydata.org/tutorial/relational.html).
> > >
> > > We are sorry for not being careful in checking the default settings when using the plotting tools.
> > >
> > > We have updated this detail in the manuscript and hope this addresses the reviewer's concern.

---

> > ### Comment · Reviewer_JmAq · 2021-11-23
> > **Longer feedback (1/2)**
> >
> > I took some time to respond to the authors because I wanted to make sure I would go over the paper again, having all the discussions and responses in mind. I do realize some of my comments/reading were incorrect, and I apologize for that; such as misunderstanding that rewards did not have to be binary. Nevertheless, I still struggle to see this paper as a paper that should be accepted.
> >
> > It still seems to me that the paper relies on four major assumptions: The problems being tackled are episodic, goal-reaching, deterministic, and that you can visit the same state multiple times. I do agree that episodic, goal-reaching tasks are somewhat common in the field, so these are not so limiting; but the assumption around determinism and that one can visit the same state multiple times is somewhat contrived. It does seem TER's performance degrades somewhat gracefully in stochastic environments (although the performance dropped by half from Figure 3(b) to F.9). More concerning is that this experiment seems to be contrived. The overall suggestion, that one could count state visitation counts is central to the argument in the Discussion, but this is a big assumption. Pseudo-counts with a dependence on density models are a whole research area exactly because of that. Arguably, if one could count state visitation counts effectively in problems in which function approximation is required, exploration would be a much easier problem.
> >
> > This difficulty to consider the idea of visiting the same state twice is the key issue for me. I grant that the method's performance does seem to decay gracefully, but in this setting it is just a much more complex solution for an ominous problem in reinforcement learning. And it is not about distractors, but it is about noise in the observations, changing observations space, etc. For example, in multiple games the score and a clock on top of the screen render it almost impossible to visit multiple states at once, mainly when you add stochasticity into the mix. So the discussion in Section 4.1 about assumptions, and the argument that Zhu et al. (2020) shows that an agent can visit repeated states even in high-dimensional state spaces is not that convincing. Atari games are deterministic and the simplest possible games that there are. The question of representation learning is central to all these issues, but it is ignored in this paper since random projections are used. Learning a representation would make the problem even harder, since it would be changing as the agent interacts with the environment, but maybe it would abstract away some of these issues.
> >
> > I still have some problems with the domains chosen for evaluation. Both the gridworld problems implemented in MiniGrid and the Sokoban instances are almost handcrafted for this type of method. Search is such an obvious solution for these methods that it is not surprising that a method that implements some sort of BFS would be effective in these environments. The problem is that I don't think these environments are representative. The only more standardized set of problems that was used, Atari games (not that they are representative, but they don't have much experimenter's bias), end up with results where TER is not beneficial.

---

> > ### Comment · Reviewer_JmAq · 2021-11-23
> > **Longer feedback (2/2)**
> >
> > Finally, I was hoping that the paper was going to be more carefully proofread and made more precise for the rebuttal, and although the claims I pointed out were fixed, there are still more. It can't be the job of a reviewer to highlight each single one of the issues of precision the paper has. I'll list some additional ones:
> >
> > - "The Q-function is trained by performing bootstrapping over states sampled from the experience replay buffer." This is an overgeneralization. There's a rich body of work in RL that doesn't use experience replay buffers, including Q-Learning.
> > - "two vertices are connected with an edge if the agent transitions between them during exploration." It seems to me there's a misunderstanding on what exploration is, since it seems every sample is added to the experience replay buffer for the graph construction.
> > - There're important references missing, such as prioritized sweeping (c.f. "Prioritized sweeping: Reinforcement learning with less data and less time"), Dyna "Dyna, an integrated architecture for learning, planning, and reacting". Sometimes wrong references are given as well, such as in Section 3, where Mnih et al. (2015) is cited instead of Lin (1992) when referring to an experience replay buffer (same happens in Section 5.1, for example). When citing TD learning, Sutton and Barto (2018) is cited, but it should have been "Learning to predict by the methods of temporal differences".
> > - In Section 3, the reward is defined to be $R(s_\tau, a_\tau)$ in the first paragraph, and then in the second paragraph, in the Q-learning update, it is written as $\mathcal{R}(s, a, s')$. The letter is different and the arguments are different. The paper needs to be proofread throughout.
> > - In Section 4.2, I believe $\phi$ is being used to refer both to the hashing function and to the feature representation.
> > - I'm not clear on the confidence intervals. Were they computed over 5 samples? 500? 100? My question in a different message about normality assumptions and bootstrapping method were not answered.
> > - I explicitly mentioned that the agent always receive a reward, at each time step, despite it possibly being zero. Section 5.2 states that "the only reward in NChain emits at.."
> >
> >
> > Because of all that, I'm still leaning towards rejection. I acknowledge I had missed some points and the assumptions made by TER are not as restrictive as I thought, and because it is shown that its performance decays gracefully, I'll increase my score to a weak reject. I also appreciate the ablations that were performed. Nevertheless, this is still a recommendation for rejecting this paper. I think the paper should do a better job at empirically convincing the reader that this approach is applicable to a wide set of environments. This should be done in two ways: different environments used for evaluation and the paper be evaluated, throughout, in the stochastic case, not making it a one paragraph extension at the end.

---

> > > ### Author Response · Authors · 2021-11-25
> > > **response to longer feedback (1/4)**
> > >
> > > We are glad that the reviewer found our response useful in clarifying our method. We would also like to express appreciation for the meticulous review and thank the reviewer for raising several points of discussion, which we believe have helped us make our submission stronger.
> > > We would like to respond to the reviewer's remaining concerns below:
> > >
> > > ### Stochastic environments
> > >
> > > We acknowledge that stochastic transition dynamics can make TER's performance worse because we use a deterministic state dependency graph. Nevertheless, we would like to clarify that the core idea of this paper is to illustrate the data-inefficiency issue of random update ordering and demonstrate that reverse update ordering mitigates this issue. This issue of update order is particularly significant in goal-reaching tasks since without first propagating rewards from goal states, the Q-function receives no useful training signals for updating. This challenge stems from the lack of sufficient learning signals and is less relevant to stochasticity. While not ideal, most current deep RL benchmarks have deterministic transitions. As such, we experimented with deterministic environments. That being said, we acknowledge that artificial stochasticity, like sticky actions, is sometimes imposed for various purposes. We show that in Figure L.15 (originally K.12) TER matches the baselines' performance, suggesting that sticky actions would not devastate TER. We believe extending TER to stochastic environments is important future work.
> > >
> > > Regarding the extension of TER in stochastic environments, we would like to clarify some points mentioned in the discussion section. Instead of estimating the pseudo counts of states, we intended to suggest that one can store the visitation count of each vertex $v$ via counting the number of states $s$ mapped to $v$ by random projection. Based on visitation counts, we compute the transition probabilities $p(v_1 \rightarrow v_2)$ for each edge. One can use $p(v_1 \rightarrow v_2)$ to sample predecessors for reverse sweeping so each predecessor will be sampled approximately based on their transition probabilities.
> > >
> > > For the comparison between Figure 3(b) and F.9, we would like to bring the reviewer's attention to that these two experiments might not be comparable. Due to stochasticity, the maximal expected return that the agent can get might be lower than that in deterministic environments. For example, in LavaCrossing-Hard, an agent could be prone to fall into lava under stochasticity because many random actions could all push the agent to lava. As such, by comparison between these two figures, it would be difficult to conclude that to what extent, the stochasticity hurts the method.

---

> > > ### Author Response · Authors · 2021-11-25
> > > **response to longer feedback (2/4)**
> > >
> > > ### Visiting the same state twice / Representation learning
> > >
> > > **Answer:**
> > >
> > > We understand the reviewer's concerns and appreciate the detailed reasoning. We would like to clarify a few points below.
> > >
> > > - In our method, observational noise only affects the graph but not the Q-function. In the worst case, when noise is large enough that two states are not mapped to the similar embeddings in the feature space, the graph would be a collection of un-connected chains. Updating the Q-values in the reverse order for these chains is known to be beneficial as shown by superior performance of EBU over Uniform/prioritized experience replay. Therefore, in deterministic environments our method outperforms state-of-the-art, whereas with stochastic noise in observations / changing observation space we will match existing state-of-the-art.
> > > - We share the reviewer's opinion that representation learning is very important. Suppose one was able to learn a good representation, such that observations obtained by adding noise / change in observation space of the same underlying "state" are mapped to the same latent vector. The question is how to use this latent space for learning the Q-function. One possibility is to learn the Q-function based on such representation or to use it an auxiliary loss aid learning of the Q-function. Our work proposes another (but complementary) way to utilize such a representation — i.e., to choose an ordering for Q-learning updates.
> > >
> > >     The reason we used random projections in our work is not that we believe that they are the best choice, but because we wanted to focus on demonstrating the importance of update ordering obtained by arranging data into a graph. Representation learning is therefore important, but complementary to our contribution. As better methods for representation learning are developed, they can be easily integrated with our work. Finally, we would like to point out that is not necessary to learn the representation for graph-building as the agent interacts with the environment. As the reviewer pointed out this might create issues, since the representation itself might change over time. However, we can also use representations from pre-trained models (although this might have its disadvantages too). E.g., many of the deep Q-learning algorithms that use a replay buffer have a burn-in period where the data collected from random exploration is used to populate the replay buffer. We can use this data to learn a representation for graph-building using contrastive learning in the style of SimCLR [1], time-contrastive learning [2], VQ-VAE [3] that learns discrete representation or other options. Note that the representation used for graph-building and learning the Q-function are separate, so using a pre-trained representation for graph-building does-not compromise the ability to fit the Q-function using interaction data. Due to several design choices and the scope of the representation learning problem, we believe addressing the representation learning is beyond the scope of this work and can be a paper in itself. In our current submission, we have chosen to perform a detailed investigation and evaluation of properties of different update ordering rather than of state representations.
> > >
> > >
> > > [1] Chen, Ting, et al. "A simple framework for contrastive learning of visual representations." *International conference on machine learning*. PMLR, 2020.
> > >
> > > [2] Savinov, Nikolay, Alexey Dosovitskiy, and Vladlen Koltun. "Semi-parametric topological memory for navigation."
> > >
> > > [3] Oord, Aaron van den, Oriol Vinyals, and Koray Kavukcuoglu. "Neural discrete representation learning."

---

> > > ### Author Response · Authors · 2021-11-25
> > > **response to longer feedback (3/4)**
> > >
> > > ### Environments
> > >
> > > **Answer:**
> > >
> > > We understand that why it might appear that problems in MiniGrid and Sokoban are handcrafted for our method. We would like to point out that specific MiniGrid[3] and Sokoban[4] environments we use are not handcrafted for our method but are part of standardized and well-cited benchmarks.
> > > Though BFS seems to be an obvious solution to accomplish tasks in MiniGrid and Sokoban, to the best of our knowledge, no prior work has shown that guiding the experience replay via BFS improves DQN performance. There is a critical distinction between directly applying BFS to solve these tasks and using BFS to guide the replay process. Directly applying BFS relies on the access to the ground truth state graph of the environment, which is not accessible in new levels of the game. Instead, our method uses BFS to guide the replay process for training the Q-value network. The learned Q-value network can generalize to new levels since it does not need the graph in the new levels.
> > > Regarding Atari, we suspect that TER is not showing gains in Atari because of task objective mismatch and difficulty on exploration. Pong and Kangaroo are not goal-reaching tasks, so TER would only match the performance of the baseline since propagating rewards from failure (e.g., missing balls) terminal states will not directly help learning. Though Freeway is a goal-reaching task, the rooms for improvement are little since the uniform experience replay already performs very well. In Venture, exploration is a major issue and it is known that commonly used $\epsilon$-greedy exploration cannot attain decent performance. Therefore, even if the Q-value update is accelerated, the absence of rewarding experience in the replay buffer would imply that the benefits of accelerated learning of Q-values are effectively washed away. Concurrent to our submission, the work of Decision Transformers [5] proposes to stitch trajectories in offline RL setup. Their methods yield improvement in minigrid, but not on ATARI. This further adds evidence that the main issue in ATARI games might be exploration and therefore, the benefits of stitching of trajectories to improve Q-learning may not be apparent.
> > >
> > > [3] [https://github.com/maximecb/gym-minigrid](https://github.com/maximecb/gym-minigrid)
> > >
> > > [4] [https://github.com/mpSchrader/gym-sokoban](https://github.com/mpSchrader/gym-sokoban)
> > >
> > > [5] Chen, Lili, et al. "Decision transformer: Reinforcement learning via sequence modeling." NeurIPS (2021).

---

> > > ### Author Response · Authors · 2021-11-25
> > > **response to longer feedback (4/4)**
> > >
> > > ### Proofread
> > >
> > > **Answer:**
> > >
> > > We appreciate the reviewer's careful reading. Sorry, that we had some imprecise statements — during the rebuttal, we were focused on providing experiments to address the concerns of the reviewers and were not able to spend as much time on revising the writing as we liked. We have updated the manuscript according to the reviewer's suggestion and promise to put substantial effort in making our arguments even more precise in the next revision. We have responses to a few comments below:
> > >
> > > - *"two vertices are connected with an edge if the agent transitions between them during exploration." It seems to me there's a misunderstanding on what exploration is, since it seems every sample is added to the experience replay buffer for the graph construction.*
> > >     - Yes, every sample (transition) is added to the reply buffer. We convert each transition to a pair of vertices $v$, $v^\prime$ respectively corresponding to the current state s and the next state $s^\prime$, and build an edge from $v$ to $v^\prime$. Note that the size of our graph will not endlessly grow since we will prune the graph when the number of transitions in the replay buffer exceeds a given threshold. The pruning implementation is detailed In Section A.5.
> > > - In Section 4.2, I believe is being used to refer both to the hashing function and to the feature representation ϕ
> > >     - We are sorry for being unclear in the use of notation. In fact, we always use $\phi$ to denote the hash function (more precisely, an encoder that transforms states to vertices. we have updated the naming of $\phi$ to vertex encoding function to reflect its meaning more precisely).
> > >
> > > Regarding the confidence interval computation, we have replied to the reviewer's original responsne.

---

> ### Comment · Reviewer_JmAq · 2021-11-29
> **Final Assessment**
>
> I've gone over the authors' rebuttal, in fact we had a couple of iterations by now. I had been harsher than I should have been in my initial assessment, and I revised my score. Nevertheless, even after the back and forth, I still think the paper is not that exciting. I'm not comfortable with the somewhat limited applicability of the proposed task. TER works quite well in an idealized setting, otherwise reverting back to previously published methods. Thus my recommendation.
>
> It seems it will be overuled by the general excitement of other reviewers, given their score. If that's the case, I recommend the authors to address my comments in the main text as that will make the paper more precise and correct.

---

### Decision · Program_Chairs · 2022-01-20

**Decision:**

Accept (Poster)

**Comment:**

A new sampling strategy for experience is proposed and compared with alternative sampling strategies. The main weakness of the paper is the limited applicability of the strategy as it only works well goal-oriented tasks, and stochasticity reduces the effectiveness. And within this setting, only good performance is shown on two gridworld-like: MiniGrid and Sokoban. In the rebuttal phase, the authors have added additional experiments that suggest applicability of the approach beyond just goal-oriented tasks, which have let several reviewers to raise their score. While general applicability of the approach is still somewhat of a concern, the authors have done enough to show the potential of their approach. Hence, I recommend acceptance.